# OTTERS: a powerful TWAS framework leveraging summary-level reference data

Qile Dai[1,2], Geyu Zhou[3], Hongyu Zhao [3,4], Urmo Võsa [5], Lude Franke [6,7], Alexis Battle [8], Alexander Teumer [9], Terho Lehtimäki [10], Olli T. Raitakari[11,12,13], Tõnu Esko[5], eQTLGen Consortium*, Michael P. Epstein [2] ✉ & Jingjing Yang [2] ✉

Most existing TWAS tools require individual-level eQTL reference data and thus are not applicable to summary-level reference eQTL datasets. The development of TWAS methods that can harness summary-level reference data is valuable to enable TWAS in broader settings and enhance power due to increased reference sample size. Thus, we develop a TWAS framework called OTTERS (Omnibus Transcriptome Test using Expression Reference Summary data) that adapts multiple polygenic risk score (PRS) methods to estimate eQTL weights from summary-level eQTL reference data and conducts an omnibus TWAS. We show that OTTERS is a practical and powerful TWAS tool by both simulations and application studies.

Transcriptome-wide association study (TWAS) is a valuable analysis strategy for identifying genes that influence complex traits and diseases through genetic regulation of gene expression[1–5]. Researchers have successfully deployed TWAS analyses to identify risk genes for complex human diseases, including Alzheimer's disease[6–8], breast cancer[9–11], ovarian cancer[12,13], and cardiovascular disease[14,15]. A typical TWAS consists of two separate stages. In Stage I, TWAS acquires individual-level genetic and expression data from relevant tissues available in a reference dataset like the Genotype-Tissue Expression (GTEx) project[16,17] or the North American Brain Expression Consortium[18], and trains multivariable regression models on the reference data treating gene expression as outcome and SNP genotype data (typically cis-SNPs nearby the test gene) as predictors to determine genetically regulated expression (GReX). After Stage I that uses the GReX regression models to estimate effect sizes of SNP predictors

that, in the broad sense, are effect sizes of expression quantitative trait loci (eQTLs), Stage II of TWAS proceeds by using these trained eQTL effect sizes to impute GReX within an independent GWAS of a complex human disease or trait. One can then test for association between the imputed GReX and phenotype, which is equivalent to a gene-based association test taking these eQTL effect sizes as corresponding test SNP weights[19–21].

For Stage I of TWAS, a variety of training tools exist for fitting GReX regression models using reference expression and genetic data, including PrediXcan[19], FUSION[20], and TIGAR[22]. While these methods all employ different techniques for model fitting, they all require individual-level reference expression and genetic data to estimate eQTL effect sizes for TWAS. Therefore, these methods cannot be applied to emerging reference summary-level eQTL results such as those generated by the eQTLGen[23] and CommonMind[24] consortia,

[1]Department of Biostatistics and Bioinformatics, Emory University School of Public Health, Atlanta, GA 30322, USA. [2]Center for Computational and Quantitative Genetics, Department of Human Genetics, Emory University School of Medicine, Atlanta, GA 30322, USA. [3]Program of Computational Biology and Bioinformatics, Yale University, New Haven, CT 06511, USA. [4]Department of Biostatistics, Yale School of Public Health, New Haven, CT 06520, USA. [5]Estonian Genome Centre, Institute of Genomics, University of Tartu, 50090 Tartu, Estonia. [6]Department of Genetics, University of Groningen, University Medical Center Groningen, 9700 RB Groningen, The Netherlands. [7]Oncode Institute, 3521 AL Utrecht, The Netherlands. [8]Department of Computer Science, and Departments of Biomedical Engineering, Johns Hopkins University, Baltimore, MD 21218, USA. [9]Institute for Community Medicine, University Medicine Greifswald, 17489 Greifswald, Germany. [10]Department of Clinical Chemistry, Fimlab Laboratories and Finnish Centre for Cardiovascular Disease Tampere, Faculty of Medicine and Health Technology, Tampere University, Tampere 33520, Finland. [11]Centre for Population Health Research, University of Turku and Turku University Hospital, 20520 Turku, Finland. [12]Research Centre of Applied and Preventive Cardiovascular Medicine, University of Turku, 20520 Turku, Finland. [13]Department of Clinical Physiology and Nuclear Medicine, Turku University Hospital, 20521 Turku, Finland. *A list of authors and their affiliations appears at the end of the paper. ✉e-mail: mpepste@emory.edu; jingjing.yang@emory.edu

which provide eQTL effect sizes and $p$ values relating individual SNPs to gene expression. The development of TWAS methods that can utilize such summary-level reference data is valuable to permit the applicability of the technique to broader analysis settings. Moreover, as TWAS power increases with increasing reference sample size[25], TWAS using summary-level reference datasets can lead to enhanced performance compared to using individual-level reference datasets since the sample sizes of the former often are considerably larger than the latter. For example, the sample size of the summary-based eQTL-Gen reference sample is 31,684 for blood, whereas the sample size of the individual-level GTEx V6 reference is only 338 for the same tissue. Consequently, TWAS analysis leveraging the summary-based eQTLGen dataset as a reference can likely provide insights into the genetic regulation of complex human traits.

In this work, we propose a framework that can use summary-level reference data to train GReX regression models required for Stage I of TWAS analysis. Our method is motivated by a variety of published polygenic risk score (PRS) methods[26–31] that can predict phenotype in a test dataset using summary-level SNP effect size estimates and $p$ values based on single SNP tests from an independent reference GWAS. We can adapt these PRS methods for TWAS since eQTL effect sizes are essentially SNP effect sizes resulting from a reference "GWAS" of gene expression. Thus, our predicted GReX in Stage II of TWAS is analogous to the PRS constructed based on training GWAS summary statistics of single SNP-trait association. Here, we adapt four representative summary-data-based PRS methods—$p$ value thresholding with linkage disequilibrium (LD) clumping (P+T)[26], frequentist LASSO[32] regression-based method lassosum[27], nonparametric Bayesian Dirichlet Process Regression (DPR) model-based[33] method SDPR[29], and Bayesian multivariable regression model-based method with continuous shrinkage (CS) priors PRS-CS[28] for TWAS analysis. We apply each of these PRS methods to first train eQTL effect sizes based on a multivariable regression model from summary-level reference eQTL data (Stage I), and subsequently use these eQTL effect sizes (i.e., eQTL weights) to impute GReX and then test GReX-trait association in an independent test GWAS (Stage II).

As we will show, the PRS method with optimal performance for TWAS depends on the underlying genetic architecture for gene expression. Since the genetic architecture of expression is unknown apriori, we maximize the performance of TWAS over different possible architectures by proposing a TWAS framework called OTTERS (Omnibus Transcriptome Test using Expression Reference Summary data). OTTERS first constructs individual TWAS tests and $p$ values using eQTL weights trained by each of the PRS techniques outlined

above, and then calculates an omnibus test $p$ value using the aggregated Cauchy association test[34] (ACAT-O) with all individual TWAS $p$ values (Fig. 1). OTTERS is applicable to both summary-level and individual-level test GWAS data within Stage II TWAS analysis.

In subsequent sections, we first describe how to use the PRS methods on summary-level reference eQTL data in Stage I TWAS, and then describe how we can use the resulting eQTL weights to perform Stage II TWAS using OTTERS. We then evaluate the performance of individual PRS methods and OTTERS using simulated expression and real genetic data based on patterns observed in real datasets. Interestingly, when we assume individual-level reference data are available, we observe that OTTERS outperforms the popular FUSION[20] approach across all simulation settings considered. Many of the individual PRS methods also outperform FUSION in these settings. We then apply OTTERS to blood eQTL summary-level data ($n = 31,684$) from the eQTLGen consortium[23] and GWAS summary data of cardiovascular disease from the UK Biobank (UKBB)[35]. By comparing OTTERS results to those of FUSION[20] using individual-level GTEx reference data of whole blood tissue, we demonstrate that OTTERS using large summary-level reference datasets and multiple gene expression imputation models can successfully reveal potential risk genes missed by FUSION based on smaller individual-level reference datasets and only one model. Finally, we conclude with a discussion.

## Results
### Method overview
For the standard two-stage TWAS approach, Stage I estimates a GReX imputation model using individual-level expression and genotype data available from a reference dataset, and then Stage II uses the eQTL effect sizes from Stage I to impute gene expression (GReX) in an independent GWAS and test for association between GReX and phenotype. GReX for test samples can be imputed from individual-level genotype data and eQTL effect size estimates. When individual-level GWAS data are not available, one can instead use summary-level GWAS data for TWAS by applying the TWAS $Z$-score statistics proposed by FUSION[20] and S-PrediXcan[36] (see details in Methods).

Since eQTL summary data are analogous to GWAS summary data where gene expression represents the phenotype, we can follow the idea from PRS methods to estimate the eQTL effect sizes based on a multivariable regression model using only marginal least squared effect estimates and $p$ values (based on a single variant test) from the eQTL summary data as well as a reference LD panel from samples of the same ancestry[26–29]. Although all PRS methods are applicable to TWAS Stage I, we only consider four representative methods—P+T[26],

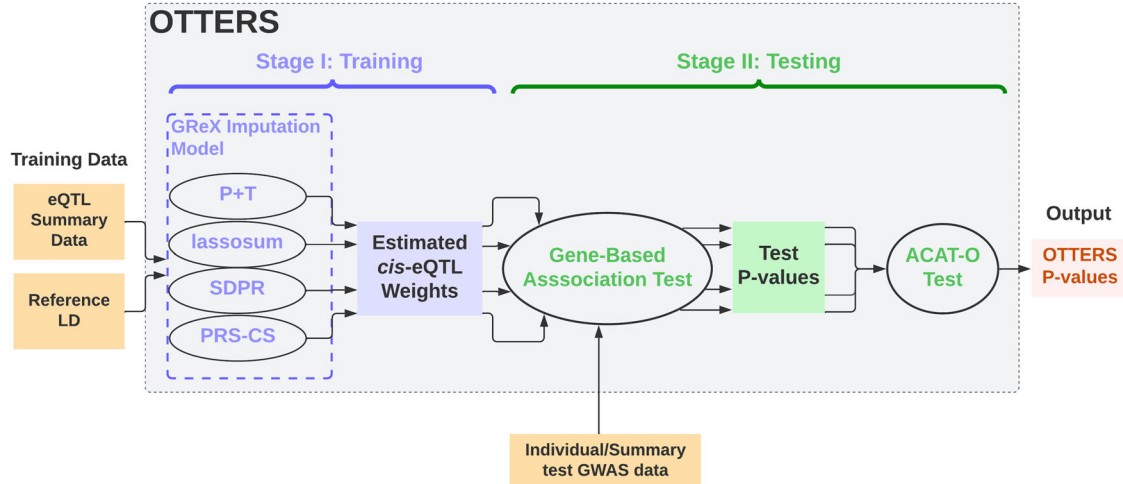

**Fig. 1 | OTTERS framework.** OTTERS estimates cis-eQTL weights from eQTL summary data and reference LD panel using four imputation models (Stage I), and conducts ACAT-O test to combine gene-based association test $p$ values from individual methods with individual/summary-level test GWAS data (Stage II).

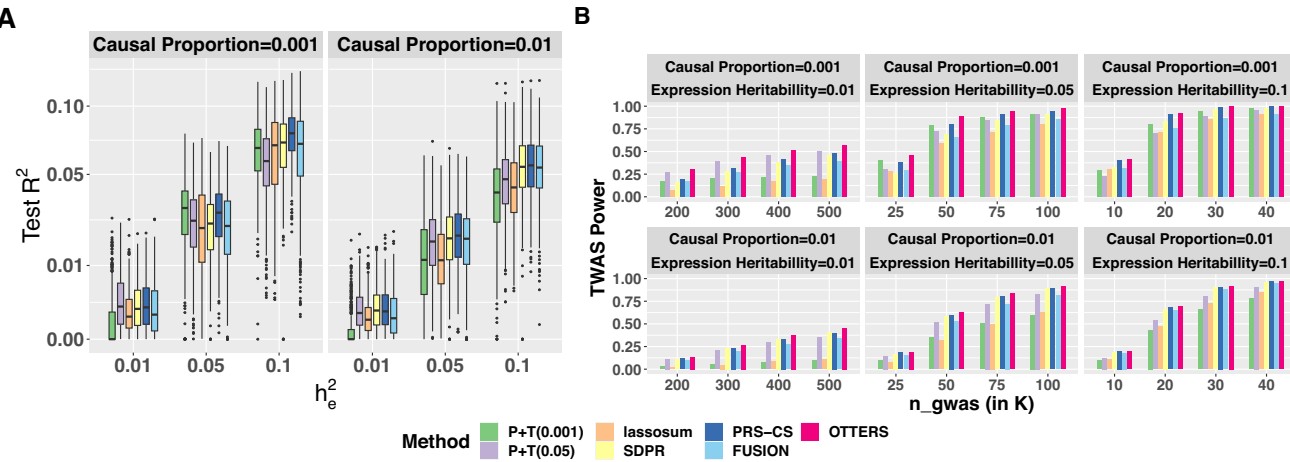

**Fig. 2 | Test $R^2$ (A) and TWAS power (B) comparison in simulation studies.** Various scenarios with proportions of true causal cis-eQTL $p_{causal} = (0.001, 0.01)$ and gene expression heritability $h_e^2 = (0.01, 0.05, 0.1)$ were considered in the simulation studies. Distribution of test $R^2$ in 5000 simulations per method per scenario was presented using box-plot (**A**). The median was shown as a black bar. The lower and upper hinges corresponded to the 25th and 75th percentiles. Whiskers extended from the hinge to the value no further than 1.5 of the interquartile range. Data beyond the end of the whiskers were plotted individually. The GWAS sample size in the x-axis of panel **B** was chosen with respect to $h_e^2$ values. The proportion of phenotype variance explained by gene expression ($h_p^2$) was set to be 0.025. TWAS was conducted using simulated GWAS Z-scores.

Frequentist lassosum[27], Nonparametric Bayesian SDPR[29], Bayesian PRS-CS[28] (see details in Methods).

As shown in Fig. 1, OTTERS first trains GReX imputation models per gene $g$ using P+T, lassosum, SDPR, and PRS-CS methods that each infers cis-eQTL weights using cis-eQTL summary data and an external LD reference panel of the same ancestry (Stage I). Once we derive cis-eQTL weights for each training method, we can impute the respective GReX using that method and perform the respective gene-based association analysis in the test GWAS dataset. We thus derive a set of TWAS $p$ values for gene $g$, one per training method. We then use these TWAS $p$ values to create an omnibus test using the ACAT-O[34] approach that employs a Cauchy distribution for inference (see details in Supplementary Methods). We refer to the $p$ value derived from ACAT-O test as the OTTERS $p$ value. The ACAT-O[34] approach has been widely used in hypothesis testing to combine multiple testing methods for the same hypothesis[37–39], which has been shown as an effective approach to leverage different test methods to increase the power while still managing to control for type I error. Adding TWAS $p$ values based on additional PRS methods to the ACAT-O test can possibly improve the power further at the cost of additional computation.

**Simulation study**

We used real genotype data from 1894 whole-genome sequencing (WGS) samples from the Religious Orders Study and Rush Memory and Aging Project (ROS/MAP) cohort[40,41] and Mount Sinai Brain Bank (MSBB) study[42] for simulation. We divided 14,772 genes into five groups according to gene length, and randomly selected 100 genes from each group (500 genes in total). We randomly split samples into 568 training (30%) and 1326 testing samples (70%) to mimic a relatively small sample size in the real reference panel for training gene expression imputation models. From the real genotype data, we simulated six scenarios with two different proportions of causal cis-eQTL, $p_{causal} = (0.001, 0.01)$, as well as three different proportions of gene expression variance explained by causal eQTL, $h_e^2 = (0.01, 0.05, 0.1)$.

We generated gene expression of gene $g$ ($\mathbf{e}_g$) using the multivariable regression model $\mathbf{e}_g = \mathbf{X}_g\mathbf{w} + \boldsymbol{\epsilon}_g$, where $\mathbf{X}_g$ represents the standardized genotype matrix of the randomly selected causal eQTL of gene $g$, $\boldsymbol{\epsilon}_g \sim N(0, (1 - h_e^2)\mathbf{I})$. We generated the eQTL effect sizes $\mathbf{w}$ from $N(0,1)$ and then re-scaled these effects to ensure that the expression variance explained by all causal variants is $h_e^2$. We generated 10 replicates of gene expression per scenario. For each simulated gene expression, we then generated 10 sets of GWAS Z-scores to perform a total of 50,000 TWAS simulations. We generated the GWAS Z-scores from a multivariate normal distribution with $\mathbf{Z} \sim MVN\left(\boldsymbol{\Sigma}_g\mathbf{w}\sqrt{n_{gwas}h_p^2}, \boldsymbol{\Sigma}_g\right)$[38], where $\mathbf{w}$ is the true causal eQTL effect sizes, $\boldsymbol{\Sigma}_g$ is the correlation matrix of the standardized genotype $\mathbf{X}_g$ from test samples, $n_{gwas}$ is the assumed GWAS sample size, and $h_p^2$ denotes the amount of phenotypic variance explained by simulated $\mathbf{GReX} = \mathbf{X}_g\mathbf{w}$ (see Methods). We set $h_p^2 = 0.025$. To calibrate power, we considered $n_{gwas} = (200K, 300K, 400K, 500K)$ for scenarios with $h_e^2 = 0.01$, $n_{gwas} = (25K, 50K, 75K, 100K)$ for scenarios with $h_e^2 = 0.05$, and $n_{gwas} = (10K, 20K, 30K, 40K)$ for scenarios with $h_e^2 = 0.1$.

In Stage I of our TWAS analysis, we applied P+T (0.001), P+T (0.05), lassosum, SDPR, and PRS-CS methods to estimate eQTL weights using eQTL summary data and the reference LD of training samples. In Stage II of the TWAS, we used the estimated eQTL weights and the simulated GWAS Z-scores to conduct a gene-based association test. In addition to gene-based association tests based on eQTL weights per training method, we further constructed the corresponding OTTERS $p$ values. We evaluated the performance of the training methods with test samples, comparing test $R^2$ that was the squared Pearson correlation coefficient between imputed GReX and simulated gene expression. We evaluated TWAS power given by the proportion of 50,000 repeated simulations with TWAS $p$ value $<2.5 \times 10^{-6}$ (genome-wide significance threshold adjusting for testing 20K independent genes).

As shown in Fig. 2, we demonstrated that the Stage I training method with optimal test $R^2$ and TWAS power depended on the underlying genetic architecture of gene expression ($p_{causal}$) as well as gene expression heritability ($h_e^2$). In situations where true cis-eQTLs were sparse ($p_{causal} = 0.001$) and the gene expression heritability was small ($h_e^2 = 0.01$), P+T (0.05) method performed the best with the highest TWAS power among all individual methods. When gene expression heritability is low ($h_e^2 = 0.01$), the power of P+T (0.001) and

lassosum methods were shown as the lowest. When gene expression heritability increased ($h_e^2 = 0.05$ or 0.1) within this sparse eQTL model, P+T (0.001) and PRS-CS were generally the optimal methods. For a less sparse model with $p_{causal} = 0.01$, SDPR and PRS-CS generally performed best among the individual methods. Relative to individual methods, we found that combining the TWAS $p$ values based on the four PRS training methods together for analysis in our OTTERS framework obtained the highest power across all scenarios.

To evaluate the type I error of the individual PRS methods along with OTTERS, we picked one simulated replicate per gene from the scenario with $h_e^2 = 0.1$ and $p_{causal} = 0.001$, simulated $2 \times 10^3$ phenotypes from $N(0,1)$, and permuted the eQTL weights for TWAS to perform a total of $10^6$ null simulations. OTTERS was shown well calibrated in the tails of the distribution as shown by quantile-quantile (Q-Q) plots of TWAS $p$ values in Supplementary Fig. S1. We also observed that OTTERS had well-controlled type I error for stringent significance levels between $10^{-4}$ and $2.5 \times 10^{-6}$ (Supplementary Table S1), which are typically utilized in TWAS. For more modest significance thresholds ($\alpha = 10^{-2}$), we noted that OTTERS had a slightly inflated type I error rate. This modest inflation is consistent with the findings of the original ACAT-O work, which showed that the Cauchy-distribution-based approximation that ACAT-O employs might not be accurate for larger $p$ values when the correlation among tests is strong[34]. This suggests that modest OTTERS $p$ values may be interpreted with caution.

We also compared the performance of our individual PRS training methods to those of FUSION, assuming individual-level reference data were available for the latter method to train GReX models. As shown in Fig. 2A, we interestingly observed that our training methods yielded similar or improved test $R^2$ compared to FUSION in this situation, with SDPR and PRS-CS outperforming FUSION across all simulation settings. Comparing TWAS power, we found that OTTERS outperformed FUSION by a considerable margin in our simulations (Fig. 2B). These simulation results suggest that, while we developed OTTERS based on PRS training methods to handle summary-level reference data, OTTERS can still improve TWAS power when individual-level reference data are available. This is likely because OTTERS accounts for multiple possible models of genetic architectures of gene expression assumed by the different PRS training methods.

## GReX imputation accuracy in GTEx V8 blood samples

To evaluate the imputation accuracy of P+T (0.001), P+T (0.05), lassosum, SDPR, and PRS-CS methods in real data, we applied these training methods to summary-level eQTL reference data from the eQTLGen consortium[23] with $n = 31,684$ blood samples, to train GReX imputation models for 16,699 genes. For test data, we downloaded the transcriptomic data of 315 blood tissue samples that are in GTEx V8 but were not part of GTEx V6 (as GTEx V6 samples contributed to the reference eQTLGen consortium summary data). For these 315 samples, we compared imputed GReX to observed expression levels. We considered trained imputation models with test $R^2 > 0.01$ as "valid" models, as suggested by previous TWAS methods[20,43]. We also compared the imputation accuracy of these five training models to those using FUSION based on a smaller individual-level training dataset (individual-level GTEx V6 reference dataset; see Methods). For such models, we compared the test $R^2$ for genes that had test $R^2 > 0.01$ by at least one training method.

We observed that PRS-CS obtained the most "valid" GReX imputation models with test $R^2 > 0.01$. Among 16,699 tested genes, PRS-CS obtained "valid" GReX imputation models for 10,337 genes, compared to 9816 genes by P+T (0.001) (5.0% less valid genes than PRS-CS), 9662 genes by P+T (0.05) (6.5% less), 8718 genes by lassosum (15.7% less), 9670 genes by SDPR (6.5% less), and 4704 genes by FUSION (54.5% less) (Table 1). Among the "valid" GReX imputation models obtained by each method, the ones trained by PRS-CS have the highest median test

$R^2$. The P+T (0.001) method obtained the second most "valid" GReX imputation models with the second largest median test $R^2$, as compared to P+T (0.05), lassosum, and SDPR (Table 1). We note that the performance of PRS-CS method was not sensitive to the global-shrinkage parameter (Supplementary Fig. S2).

By comparing test $R^2$ per "valid" GReX imputation model by PRS-CS versus the other methods (Fig. 3), we observed that PRS-CS had the best overall performance for imputing GReX as it provided the most "valid" models with higher GReX imputation accuracy compared to P+T methods, lassosum, SDPR, and FUSION. Comparing the test $R^2$ among the other four training methods, we observed that these two P+T methods obtained similar test $R^2$ per "valid" model. Meanwhile, the test $R^2$ per valid model varied widely among the P+T methods, lassosum, and SDPR (Supplementary Fig. S3), suggesting that none of these four were optimal across all genes and their performance likely depended on the underlying unknown genetic architecture. These results are consistent with our simulation results.

## TWAS of cardiovascular disease

Using the eQTL weights trained by P+T (0.001), P+T (0.05), lassosum, SDPR, and PRS-CS methods with the eQTLGen[23] reference data and reference LD from GTEx V8 WGS samples[44], we applied our OTTERS framework to the summary-level GWAS data of Cardiovascular Disease from UKBB ($n = 459,324$, case fraction = 0.319)[35] (see Methods). We performed TWAS of cardiovascular disease for 16,678 genes. First, for each gene, we obtained TWAS $p$ values per individual training method (P+T (0.001), P+T (0.05), lassosum, SDPR, and PRS-CS). Second, we performed genomic control[45] for TWAS test statistics generated under each specific training model, by scaling all test statistics to ensure that the median test $p$ value equals to 0.5. Last, we only considered genes with test GReX $R^2 > 0.01$ by at least one PRS training method in additional GTEx V8 samples in the follow-up ACAT-O test. We combined the adjusted $p$ values across all five training models using ACAT-O to obtain our OTTERS test statistics and $p$ values. Genes with OTTERS $p$ values $< 2.998 \times 10^{-6}$ (Bonferroni corrected significance level) were identified as significant TWAS genes for cardiovascular risk.

In total, we identified 40 significant TWAS genes by using OTTERS. To identify independently significant TWAS genes, we calculated the $R^2$ (squared Pearson correlation) between the GReX predicted by PRS-CS for each pair of genes. For a pair of genes with the predicted GReX $R^2 > 0.5$, we only kept the gene with the smaller TWAS $p$ value as the independently significant gene. OTTERS obtained 38 independently significant TWAS genes (Table 2 and Fig. 3B), compared to 17 independently significant genes by P+T (0.001), 11 by P+T (0.05), 10 by lassosum, 41 by SDPR, and 12 by PRS-CS. Among these 38 independent TWAS risk genes identified by OTTERS, gene *RP11-378A13.1* (OTTERS $p$ value = $9.78 \times 10^{-9}$) was not within 1 MB of any known GWAS risk loci with genomic-control corrected $p$ value $< 5 \times 10^{-8}$ in the UKBB summary-level GWAS data. This gene *RP11-378A13.1* was also identified to be a significant TWAS risk gene in blood tissue for systolic blood pressure, high cholesterol, and cardiovascular disease by FUSION[1].

We compared our OTTERS results with the TWAS results shown on TWAS hub (see Data availability) obtained by FUSION using the same UKBB GWAS summary data of cardiovascular disease but using a smaller individual-level reference expression dataset from GTEx V6 (whole blood tissue, $n = 338$). Of the 38 independent genes that OTTERS identified from TWAS with eQTLGen reference data of whole blood, FUSION only identified 8 of these genes (*CLCN6, PSRC1, RP11-378A13.1, CAMK1D, SIDT2, MTHFSD, NTN5, OPRL1*) when using the GTEx V6 reference data of the same tissue. FUSION did identify 13 additional OTTERS genes (*NPPA, CPEB4, NT5C2, TNNT3, C11orf49, CSK, FES, MBTPS1, ACE, MRI1, HAUS8, RPL28, CTSZ*), when considering all available tissue types in GTEx V6 reference data. These genes were identified by FUSION when considering the GTEx V6 reference data of artery, thyroid, adipose visceral, nerve tibial tissues, etc. For example, the

**Table 1 | Test $R^2$ in $n = 315$ whole blood tissue samples from GTEx V8**

| | P+T (0.001) | P+T (0.05) | lassosum | SDPR | PRS-CS | FUSION[b] |
|---|---|---|---|---|---|---|
| No. of genes with $R^2 > 0.01$ | 9816 | 9662 | 8718 | 9670 | 10,337 | 4704 |
| Median $R^{2a}$ | 0.0440 | 0.0430 | 0.0416 | 0.0418 | 0.0517 | 0.0367 |

[a]Median $R^2$ among genes with test $R^2 > 0.01$ per method.
[b]FUSION was trained on GTEx V6 blood samples, while all other training methods were trained using eQTLGen summary statistics ($n = 31,684$) and reference LD from GTEx V8 samples.

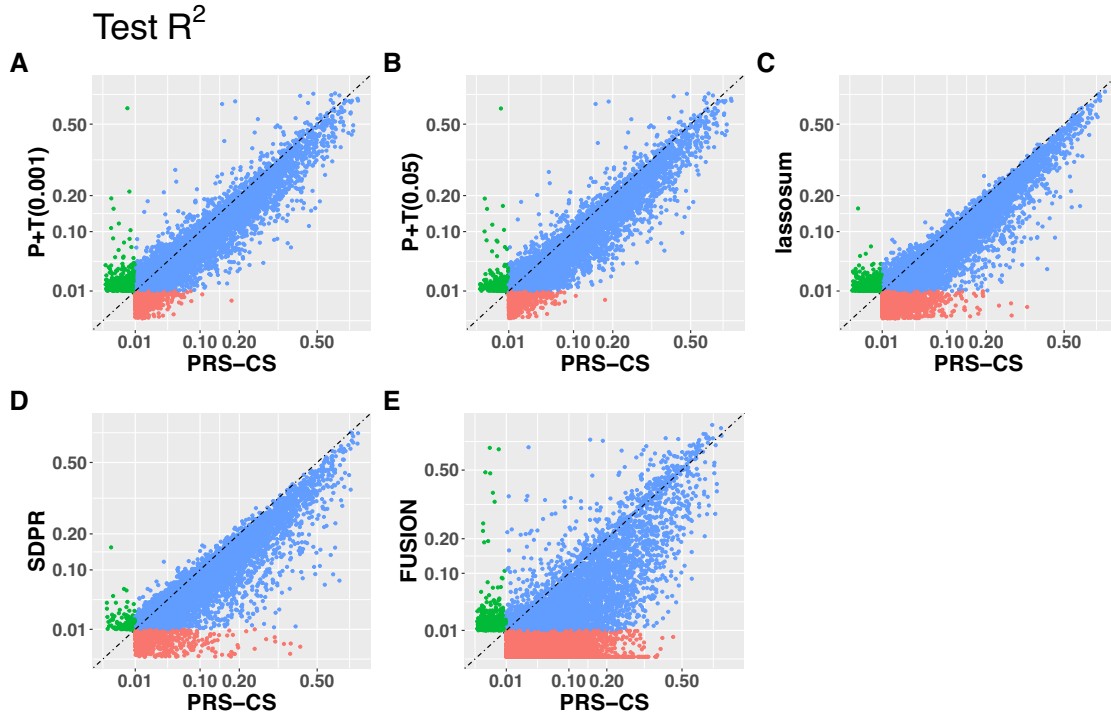

**Fig. 3 | Test $R^2$ by PRS-CS versus P+T (0.001), P+T (0.05), lassosum, SDPR and FUSION.** Test $R^2$ by PRS-CS versus P+T (0.001) (**A**), P+T (0.05) (**B**), lassosum (**C**), SDPR (**D**), and FUSION (**E**) with 315 GTEx V8 test samples, with different colors denoting whether test $R^2 > 0.01$ only by PRS-CS (red), only by the *y*-axis method (green), or both methods (blue). Genes with test $R^2 > 0.01$ by at least one method were included in the plot.

most significant gene *FES* (OTTERS *p* value = $2.87 \times 10^{-32}$) was identified by FUSION using GTEx reference data of artery tibial, thyroid, and adipose visceral omentum tissues, and was also identified as a TWAS risk gene for high blood pressure, which is strongly related to cardiovascular disease[46].

Our OTTERS method also identified 17 genes (*LINC01093, SER-PINB6, CARMIL1, ZSCAN12P1, HCG4P7, HCG4P3, HLA-S, PSPHP1, LPL, PTP4A3, SLCO3A1, RALBP1, SULT2B1, EDN3, ZBTB46, FAM3B, MX1*) that were not detected by FUSION using GTEx V6 data, where *EDN3* (Endothelin 3, a member of the endothelin family) was shown to be active in the cardiovascular system and play an important role in the maintenance of blood pressure or generation of hypertension[47].

By comparing OTTERS results with the ones obtained by individual methods (Table 2, Fig. 4 and Supplementary Fig. S4), we found that all individual methods contributed to the OTTERS results. For example, the gene *LINC01093* was only identified by lassosum, while genes *CPEB4, SIDT2*, and *ACE* were only detected by PRS-CS and SDPR and the gene *EDN3* was only identified by the P+T methods. To better understand the differences among individual methods, we plotted the eQTL weights estimated by P+T (0.001), P+T (0.05), lassosum, SDPR, and PRS-CS for three example genes that were only detected by one or two individual methods (Supplementary Figs. S5–S7). For these genes, we plotted the eQTL weights produced by each method with such weights color coded with respect to $-\log_{10}$(GWAS *p* values) from the UKBB GWAS summary statistics and shape coded with respect to the

direction of UKBB GWAS *Z*-score statistics. Generally, significant TWAS *p* values would be obtained by methods that obtained eQTL weights with relatively large magnitudes for SNPs with relatively more significant GWAS *p* values.

In Supplementary Fig. S5, we showed the eQTL weights for gene *SIDT2*, which was a significant risk gene identified by both PRS-CS and SDPR, and had *p* values < $10^{-4}$ by other methods. Compared to lassosum, SDPR had more significant GWAS SNPs colocalized with eQTLs having relatively large weights in the test region, and PRS-CS had more non-significant GWAS SNPs colocalized with eQTLs having zero weights. Compared to the P+T methods, SDPR and PRS-CS based on a multivariate regression model modeled LD among all test SNPs, and thus estimated eQTL weights leading to significant TWAS findings. In Supplementary Fig. S6, we provided the results of gene *EDN3*, which was only identified by P+T methods (*p* values ≤ $9.15 \times 10^{-8}$). Compared to P+T methods, SDPR (*p* value = $5.9 \times 10^{-3}$) and PRS-CS (*p* value = 0.0158) had fewer significant GWAS SNPs colocalized with eQTLs that had relatively large weights in the test region, while lassosum (*p* value = $8.6 \times 10^{-6}$) assigned relatively large weights to more non-significant GWAS SNPs. In Supplementary Fig. S7, we provided results for gene *LINC01093*, which was only identified by lassosum. For this gene, SDPR and PRS-CS estimated near-zero weights for most test SNPs with significant GWAS *p* values in the test region. Most significant GWAS SNPs did not have eQTL test *p* values < 0.001 or 0.05, and were thus filtered out by P+T methods. lassosum was the only method that produced

**Table 2 | Independent TWAS risk genes of cardiovascular disease identified by OTTERS**

| CHROM | ID | OTTERS | P+T (0.001) | P+T (0.05) | lassosum | SDPR | PRS-CS |
|---|---|---|---|---|---|---|---|
| 1 | CLCN6[a] | **5.75E−15** | **4.94E−09** | **5.40E−08** | **8.77E−09** | **1.19E−15** | **1.43E−09** |
| 1 | NPPA[b] | **4.32E−08** | **1.55E−08** | **2.14E−07** | – | – | 6.71E−06 |
| 1 | PSRC1[a] | **8.37E−20** | **5.68E−08** | **8.46E−07** | **6.26E−11** | **1.67E−20** | **1.41E−12** |
| 2 | RP11-378A13.1[a] | **9.78E−09** | 3.97E−02 | 4.98E−02 | 1.62E−05 | **1.96E−09** | 1.15E−04 |
| 4 | LINC01093[c] | **2.57E−09** | 9.85E−02 | 5.31E−02 | **5.13E−10** | 1.08E−02 | 2.41E−02 |
| 5 | CPEB4[b] | **3.05E−14** | 1.26E−02 | 2.05E−02 | 2.70E−05 | **6.05E−15** | **1.60E−07** |
| 6 | SERPINB6[c] | **1.47E−07** | 2.12E−01 | 2.24E−01 | 7.56E−03 | **2.95E−08** | 7.53E−04 |
| 6 | CARMIL1[c] | **9.23E−09** | 5.34E−03 | 3.41E−03 | 4.15E−03 | **1.85E−09** | 1.72E−03 |
| 6 | ZSCAN12P1[c] | **1.84E−08** | 6.00E−01 | 5.75E−01 | 4.62E−01 | **3.67E−09** | 3.10E−01 |
| 6 | HCG4P7[c] | **8.93E−50** | 3.70E−01 | 3.69E−01 | 2.30E−01 | **1.79E−50** | 7.26E−01 |
| 6 | HCG4P3[c] | **5.33E−20** | 4.20E−01 | 4.05E−01 | 5.03E−04 | **1.07E−20** | 2.42E−03 |
| 6 | HLA-S[c] | **4.57E−07** | 7.13E−01 | 7.31E−01 | 3.02E−01 | **9.14E−08** | 2.33E−01 |
| 7 | PSPHP1[c] | **1.21E−09** | 2.17E−01 | 2.26E−01 | 9.65E−03 | **2.43E−10** | 1.10E−01 |
| 8 | LPL[c] | **5.73E−07** | 1.78E−03 | 3.26E−03 | 4.44E−02 | **1.15E−07** | 1.05E−04 |
| 8 | PTP4A3[c] | **1.28E−06** | 8.13E−02 | 8.33E−02 | 6.23E−05 | **2.58E−07** | 1.67E−03 |
| 10 | CAMK1D[a] | **2.51E−09** | 3.83E−02 | 4.97E−02 | 1.23E−03 | **5.03E−10** | 4.97E−05 |
| 10 | NT5C2[b] | **1.21E−07** | **1.69E−06** | **2.92E−06** | 1.64E−05 | **3.15E−07** | **2.69E−08** |
| 11 | TNNT3[b] | **1.67E−10** | **1.09E−06** | 3.33E−06 | **2.03E−09** | **3.40E−11** | **4.01E−07** |
| 11 | C11orf49[b] | **2.28E−06** | **8.55E−07** | **1.78E−06** | 5.44E−05 | – | 2.93E−04 |
| 11 | SIDT2[a] | **7.26E−09** | 6.14E−05 | 1.33E−04 | 3.66E−05 | **1.46E−09** | **3.81E−07** |
| 15 | CSK[b] | **2.30E−09** | **1.70E−07** | **2.15E−06** | **7.41E−10** | **2.80E−09** | **2.17E−09** |
| 15 | FES[b] | **2.87E−32** | **4.78E−08** | **1.23E−06** | **9.13E−24** | **5.75E−33** | **1.94E−15** |
| 15 | SLCO3A1[c] | **3.78E−08** | 1.85E−02 | 3.15E−02 | 4.65E−05 | **7.57E−09** | 1.14E−03 |
| 16 | MBTPS1[b] | **5.80E−08** | 2.62E−01 | 3.05E−01 | 9.15E−04 | **1.16E−08** | 2.34E−03 |
| 16 | MTHFSD[a] | **4.65E−07** | 5.16E−02 | 5.94E−02 | 1.65E−02 | **9.30E−08** | 3.20E−03 |
| 17 | ACE[b] | **9.42E−07** | **4.93E−06** | **1.03E−05** | **4.23E−06** | **9.66E−07** | **2.68E−07** |
| 18 | RALBP1[c] | **1.40E−06** | 1.48E−01 | 1.54E−01 | 2.12E−04 | **2.81E−07** | 5.55E−03 |
| 19 | MRI1[b] | **8.38E−09** | 8.34E−03 | 1.60E−02 | 7.79E−03 | **1.68E−09** | 2.65E−03 |
| 19 | HAUS8[b] | **1.60E−07** | **4.41E−08** | **1.38E−07** | **1.67E−06** | **1.42E−06** | 3.29E−05 |
| 19 | SULT2B1[c] | **2.32E−06** | **7.73E−07** | – | – | 2.97E−02 | 1.10E−02 |
| 19 | NTN5[a] | **9.03E−10** | **2.75E−08** | **1.16E−07** | 6.23E−06 | **1.85E−10** | **9.73E−09** |
| 19 | RPL28[b] | **3.76E−07** | 7.33E−02 | 1.16E−01 | 6.64E−03 | **7.52E−08** | 4.23E−03 |
| 20 | CTSZ[b] | **3.32E−09** | 2.57E−02 | 1.99E−02 | **3.40E−09** | **8.25E−10** | 1.04E−01 |
| 20 | EDN3[c] | **1.29E−07** | **3.61E−08** | **9.15E−08** | 8.60E−06 | 5.90E−03 | 1.58E−02 |
| 20 | ZBTB46[c] | **1.07E−06** | **2.83E−07** | 8.35E−06 | – | 1.81E−03 | **1.27E−05** |
| 20 | OPRL1[a] | **5.84E−07** | **3.44E−07** | **2.69E−06** | 1.85E−03 | 5.51E−05 | **1.90E−07** |
| 21 | FAM3B[c] | **1.08E−10** | 2.28E−02 | 2.58E−02 | 8.07E−06 | **2.17E−11** | 1.04E−05 |
| 21 | MX1[c] | **6.04E−22** | 4.36E−01 | 3.83E−01 | **3.16E−07** | **1.21E−22** | 1.24E−03 |

Reference eQTL summary data from eQTLGen consortium and GWAS summary data from UKBB were used. The corresponding TWAS $p$ values by 5 individual PRS methods and OTTERS are shown in the table with significant $p$ values $< 2.998 \times 10^{-6}$ (Bonferroni corrected significance level) in bold, and those for genes with test GReX $R^2 \leq 0.01$ are shown as a dash. $p$ values were the genomic-control corrected $p$ values from the $Z$-score test from TWAS (two-sided).
[a]Risk gene of UKBB cardiovascular disease in TWAS-hub identified using GTEx whole blood tissue.
[b]Risk genes of UKBB cardiovascular disease in TWAS-hub identified using other GTEx tissue types.
[c]Risk gene of UKBB cardiovascular disease not in TWAS-hub.

relatively large eQTL weights that colocalized with GWAS-significant SNPs.

These results were consistent with our simulation study results, demonstrating that the performance of different individual methods depended on the underlying genetic architecture. We do note that there were a handful of genes identified by an individual method that were not significant using OTTERS (Supplementary Table S2). Nonetheless, the omnibus test borrows strength across all individual methods, thus generally achieving higher TWAS power and identifying the group of most robust TWAS risk genes.

By examining the Q-Q plots of TWAS $p$ values, we observed moderate inflation for all methods (Supplementary Fig. S8). Such inflation in TWAS results is not uncommon[48–50], which could be due to similar inflation in the GWAS summary data and not distinguishing the pleiotropy and mediation effects for considered gene expression and phenotype of interest[51] (Supplementary Fig. S9). We also observed a notable inflation in the GWAS $p$ values of cardiovascular disease from UKBB (Supplementary Fig. S9), as we estimated the LD score regression[52] intercept to be 1.1 from the GWAS summary data.

We did not consider directly comparing to FUSION in our above TWAS analyses of cardiovascular disease since we used the summary-level reference data eQTLGen. However, to assess the performance of OTTERS and FUSION in a real study where individual-level reference data are available, we performed an additional TWAS analysis of

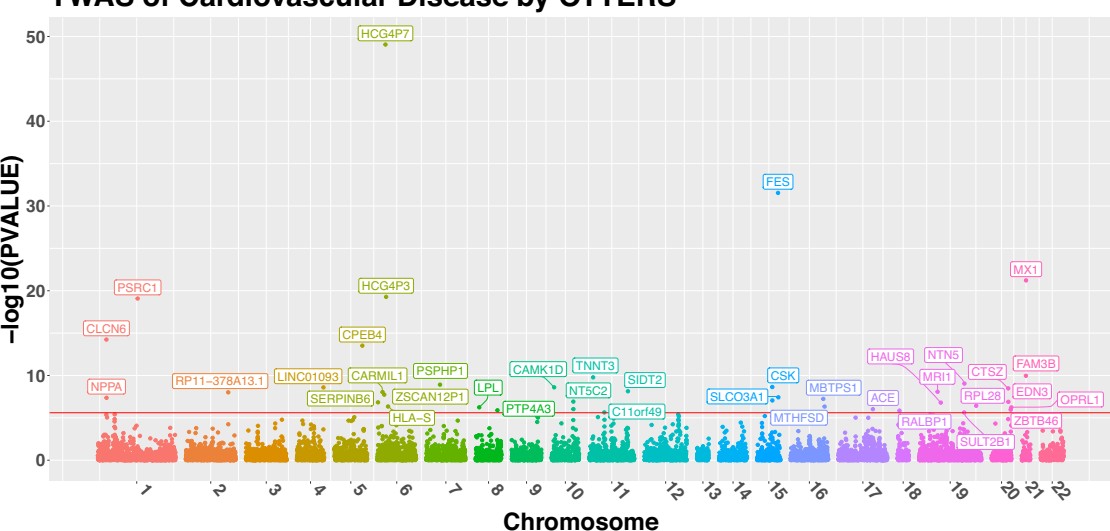

**Fig. 4 | Manhattan plot of TWAS results by OTTERS.** Manhattan plot of TWAS results by OTTERS using GWAS summary-level statistics of cardiovascular disease and imputation models fitted based on eQTLGen summary statistics. The *x*-axis represented the genomic position, and the *y*-axis represented −$\log_{10}$(*p* values). *p* values were the genomic-control corrected *p* values from the *Z*-score test from TWAS (two-sided). Independently significant TWAS risk genes were labeled.

cardiovascular disease in the UK Biobank using the GTEx V8 data of 574 whole blood samples as the reference data. We trained OTTERS Stage I using cis-eQTL summary statistics obtained from these 574 GTEx V8 whole blood samples and reference LD from GTEx V8 WGS samples, and trained FUSION models using individual-level genotype data and gene expression data of the same 574 whole blood samples.

We tested TWAS association for 19,653 genes and identified genes with TWAS *p* values < $2.53 \times 10^{-6}$ (Bonferroni corrected significance level) as significant TWAS genes. Training $R^2 > 0.01$ was used to select "valid" GReX imputation models for TWAS (Supplementary Fig. S10). To identify independently significant TWAS genes, we calculated the training $R^2$ between the GReX predicted by lassosum for of each pair of genes, since lassosum had the best training $R^2$ (Supplementary Fig. S10). For a pair of genes with the predicted GReX $R^2 > 0.5$, we only kept the gene with the smaller TWAS *p* value as the independently significant gene. As a result, OTTERS obtained 34 independently significant TWAS genes, while FUSION identified 21 independently significant TWAS genes (Supplementary Fig. S11). A total of 14 genes were identified by both FUSION and OTTERS (Supplementary Table S3).

These results demonstrate the advantages of OTTERS for using multiple PRS training methods to account for the unknown genetic architecture of gene expression, which is consistent in our simulation results. These results also showed the advantage of using eQTL summary data with a larger training sample size, as more independently significant TWAS genes were identified by using the eQTLGen summary reference data (38 vs. 34), even with a more stringent rule (test instead of training $R^2 > 0.01$) applied to select test genes with "valid" GReX imputation models.

**Computational time**

The computational time per gene of different PRS methods depends on the number of test variants considered for the target gene. Thus, we calculated the computational time and memory usage for four groups of genes whose test variants were <2000, between 2000 and 3000, between 3000 and 4000, and >4000, respectively. Among all tested genes in our real studies, the median number of test variants per gene is 3152, and the proportion of genes in each group is 10.3%, 33.4%, 34.5%, and 21.8%, respectively. For each group, we randomly selected ten genes on Chromosome 4 to evaluate the average computational

time and memory usage per gene. We benchmarked the computational time and memory usage of each method on one Intel(R) Xeon(R) processor (2.10 GHz). The evaluation was based on 1000 MCMC iterations for SDPR and PRS-CS (default) without parallel computation (Supplementary Table S4). We showed that P+T and lassosum were computationally more efficient than SDPR and PRS-CS, whose speeds were impeded by the need of MCMC iterations. Between the two Bayesian methods, SDPR implemented in C++ uses significantly less time and memory than PRS-CS implemented in Python.

**Discussion**

Our OTTERS framework represents an omnibus TWAS tool that can leverage summary-level expression and genotype results from a reference sample, thereby robustly expanding the use of TWAS into more settings. To this end, we adapted and evaluated five different PRS methods assuming different underlying genetic models, including the relatively simple method P+T[26] with two different *p* value thresholds (0.001 and 0.05), the frequentist method lassosum[27], as well as the Bayesian methods PRS-CS[28] and SDPR[29] within our omnibus test for optimal inference. We note that additional PRS methods such as MegaPRS[30] or PUMAS[31] could also be implemented as additional OTTERS Stage I training methods. Higher TWAS power might be obtained by adding more PRS methods in OTTERS Stage I, with additional computation cost. We also note that the existing SMR-HEIDI[53] method, which uses summary-level data from GWAS and eQTL studies to test for possible causal genetic effects of a trait of interest that were mediated through gene expression, could also be used as an alternative method besides TWAS. However, the SMR method generally restricts eQTL for consideration, excluding those where the eQTL *p* values larger than a certain threshold, e.g., 0.05.

In simulation studies, we demonstrated that the performance of each of these five PRS methods depended substantially on the underlying genetic architecture for gene expression, with P+T methods generally performing better for sparse architecture, whereas the Bayesian methods performing better for denser architecture. Consequently, since the genetic architecture of gene expression is unknown apriori, we believe this justifies the use of the omnibus TWAS test implemented in OTTERS for practical use, as this test had near-optimal performance across all simulation scenarios considered. While we

developed our methods with summary-level reference data in mind, we note that our prediction methods and OTTERS perform well (in terms of imputation accuracy and power) relative to existing TWAS methods like FUSION when individual-level reference data are available.

In our real data application using UKBB GWAS summary-level data, we compared OTTERS TWAS results using reference eQTL summary data from eQTLGen consortium to FUSION TWAS results using a substantially smaller individual-level reference dataset from GTEx V6. OTTERS identified 13 significant TWAS risk genes that were missed by FUSION using individual-level GTEx V6 reference data of blood tissue, suggesting that the use of larger reference datasets like eQTLGen in TWAS can provide additional findings. Interestingly, the genes missed by FUSION were instead detected using individual-level GTEx reference data of other tissue types that are more directly related to cardiovascular disease. By comparing OTTERS to FUSION when the same individual-level GTEx V8 reference data of whole blood samples were used, we still observed that OTTERS identified more risk genes than FUSION, which we believe is due to the former method accounting for the unknown genetic architecture of gene expression by using multiple regression methods to train GReX imputation models. These applied results were consistent with our simulation results.

Among all individual methods, P+T is the most computationally efficient method. The Bayesian methods SDPR and PRS-CS require more computation time than the frequentist method lassosum as the former set of methods require a large number of MCMC iterations for model fit. By comparing the performance of these five methods in terms of the imputation accuracy and TWAS power in simulations and real applications, we conclude that none of these methods was optimal across different genetic architectures. We found that all methods provided distinct and considerable contributions to the final OTTERS TWAS results. These results demonstrate the benefits of OTTERS in practice, since OTTERS can combine the strength of these individual methods to achieve optimal performance.

To enable the use of OTTERS by the public, we provide an integrated tool (see Code Availability) to (1) Train GReX imputation models (i.e., estimate eQTL weights in Stage I) using eQTL summary data by P+T, lassosum, SDPR, and PRS-CS; (2) Conduct TWAS (i.e., testing gene-trait association in Stage II) using both individual-level and summary-level GWAS data with the estimated eQTL weights; and (3) Apply ACAT-O to aggregate the TWAS $p$ values from individual training methods. Since the existing tools for P+T, lassosum, SDPR, and PRS-CS were originally developed for PRS calculations, we adapted and optimized them for training GReX imputation models in our OTTERS tool. For example, we integrate TABIX[54] and PLINK[55] tools in OTTERS to extract input data per target gene more efficiently. We also enable parallel computation in OTTERS for training GReX imputation models and testing gene-trait association of multiple genes.

The OTTERS framework does have its limitations. First, training GReX imputation models by all individual methods on average cost ~20 min for all five training models per gene, which might be computationally challenging for studying eQTL summary data of multiple tissue types and for ~20K genome-wide genes. Users might consider prioritizing P+T (0.001), lassosum, and SDPR training methods, as these three provide complementary results in our studies. Second, the currently available eQTL summary statistics are mainly derived from individuals of European descent. Our OTTERS trained GReX imputations model based on these eQTL summary statistics, and the resulting imputed GReX could consequently have attenuated cross-population predictive performance[56]. This might limit the transferability of our TWAS results across populations. Third, our OTTERS cannot provide the direction of the identified gene-phenotype associations, which should be referred to as the sign of the TWAS $Z$-score statistic per training method. Last, even though the method applies to integrate

both cis- and trans- eQTL with GWAS data, the computation time and availability of summary-level trans-eQTL reference data are still the main obstacles. Our current OTTERS tool only considers cis-eQTL effects. Extension of OTTERS to enable cross-population TWAS and incorporation of trans-eQTL effects is part of our ongoing research but is out of the scope of this work.

Our OTTERS framework using large-scale eQTL summary data has the potential to identify more significant TWAS risk genes than standard TWAS tools that use smaller individual-level reference transcriptomic data and use only a single regression method for training GReX imputation models. This tool provides the opportunity to leverage not only available public eQTL summary data of various tissues for conducting TWAS of complex traits and diseases, but also the emerging summary-level data of other types of molecular QTL such as splicing QTLs, methylation QTLs, metabolomics QTLs, and protein QTLs. For example, OTTERS could be applied to perform proteome-wide association studies using summary-level reference data of genetic-protein relationships such as those reported by the SCAL-LOP consortium[57], and epigenome-wide association studies using summary-level reference data of methylation-phenotype relationships reported by Genetics of DNA Methylation Consortium (GoDMC) (see Data availability). OTTERS would be most useful for broad researchers who only have access to summary-level QTL reference data and summary-level GWAS data. The feasibility of integrating summary-level molecular QTL data and GWAS data makes our OTTERS tool valuable for wide application in current multi-omics studies of complex traits and diseases.

## Methods

### Traditional two-stage TWAS analysis

Stage I of TWAS estimates a GReX imputation model using individual-level expression and genotype data available from a reference dataset. Consider the following GReX imputation model from $n$ individuals and $m$ SNPs (multivariable regression model assuming linear additive genetic effects) within the reference dataset:

$$\mathbf{e}_g = \mathbf{X}_g \mathbf{w} + \boldsymbol{\epsilon}_g, \boldsymbol{\epsilon}_g \sim N(0, \sigma_\epsilon^2 \mathbf{I}). \qquad (1)$$

Here, $\mathbf{e}_g$ is a vector representing gene expression levels of gene $g$, $\mathbf{X}_g$ is an $n \times m$ matrix of genotype data of SNP predictors proximal or within gene $g$, $\mathbf{w}$ is a vector of genetic effect sizes (referred to as a broad sense of eQTL effect sizes), and $\boldsymbol{\epsilon}_g$ is the error term. Here, we consider only cis-SNPs within 1 MB of the flanking 5' and 3' ends as genotype predictors that are coded within $\mathbf{X}_g$[19,20,22]. Once we configure the model in Eq. (1), we can employ methods like PrediXcan, FUSION, and TIGAR to fit the model and obtain estimates of eQTL effect sizes ($\hat{\mathbf{w}}$).

Stage II of TWAS uses the eQTL effect sizes ($\hat{\mathbf{w}}$) from Stage I to impute gene expression (GReX) in an independent GWAS and then test for association between GReX and phenotype. Given individual-level GWAS data with genotype data $\mathbf{X}_{new}$ and eQTL effect sizes ($\hat{\mathbf{w}}$) from Stage I, the GReX for $\mathbf{X}_{new}$ can be imputed by $\widehat{\mathbf{GReX}} = \mathbf{X}_{new}\hat{\mathbf{w}}$. The follow-up TWAS would test the association between $\widehat{\mathbf{GReX}}$ and phenotype $\mathbf{y}$ based on a generalized linear regression model, which is equivalent to a gene-based association test taking $\hat{\mathbf{w}}$ as test SNP weights. When individual-level GWAS data are not available, one can apply FUSION and S-PrediXcan test statistics to summary-level GWAS data as follows:

$$Z_{g,FUSION} = \frac{\sum_{j=1}^{J}(\hat{w}_j Z_j)}{\sqrt{\hat{\mathbf{w}}' \mathbf{V} \hat{\mathbf{w}}}}, Z_{g,S-PrediXcan} = \frac{\sum_{j=1}^{J}(\hat{w}_j \hat{\sigma}_j Z_j)}{\sqrt{\hat{\mathbf{w}}' \mathbf{V} \hat{\mathbf{w}}}} \qquad (2)$$

where $Z_j$ is the single variant $Z$-score test statistic in GWAS for the $j$th SNP, $j = 1, \ldots, J$, for all test SNPs that have both eQTL weights with respect to the test gene $g$ and GWAS $Z$-scores; $\hat{\sigma}_j$ is the genotype

standard deviation of the $j$th SNP; and $\mathbf{V}$ denotes the genotype correlation matrix in FUSION $Z$-score statistic and genotype covariance matrix in S-PrediXcan $Z$-score statistic of the test SNPs. In particular, $\hat{\sigma}_j$ and $\mathbf{V}$ can be approximated from a reference panel with genotype data of samples of the same ancestry such as those available from the 1000 Genomes Project[58]. If $\hat{\mathbf{w}}$ are standardized effect sizes estimated assuming standardized genotype $\mathbf{X}_g$ and gene expression $\mathbf{e}_g$ in Eq. (1), FUSION and S-PrediXcan $Z$-score statistics are equivalent[13]. Otherwise, the S-PrediXcan $Z$-score should be applied to avoid false-positive inflation.

## TWAS Stage I analysis using summary-level reference data

We now consider a variation of TWAS Stage I to estimate cis-eQTL effect sizes $\hat{\mathbf{w}}$ based on a multivariable regression model (Eq. (1)) from summary-level reference data. We assume that the summary-level reference data provide information on the association between a single genetic variant $j$ ($j = 1, \ldots, m$) and expression of gene $g$. This information generally consists of effect size estimates ($\widetilde{w}_j, j = 1, \ldots, m$) and $p$ values derived from the following single variant regression models:

$$\mathbf{e}_g = \mathbf{x}_j w_j + \boldsymbol{\epsilon}_j, \boldsymbol{\epsilon}_j \sim N\left(0, \sigma_{\epsilon_j}^2 \mathbf{I}\right), j = 1, \ldots, m. \quad (3)$$

Here, $\mathbf{x}_j$ is an $n \times 1$ vector of genotype data for genetic variant $j$, and $\boldsymbol{\epsilon}_j$ is the error term. Since eQTL summary data are analogous to GWAS summary data where gene expression represents the phenotype, we can estimate the eQTL effect sizes $\hat{\mathbf{w}}$ using marginal least squared effect estimates ($\widetilde{w}_j, j = 1, \ldots, m$) and $p$ values from the QTL summary data as well as reference LD information of the same ancestry[26–29]. Although all PRS methods apply to the TWAS Stage I framework, we only consider four representative methods as follows:

P+T: the P+T method selects eQTL weights by LD clumping and $p$ value Thresholding[26]. Given threshold $P_T$ for $p$ values and threshold $R_T$ for LD $R^2$, we first exclude SNPs with marginal $p$ values from eQTL summary data greater than $P_T$ or strongly correlated (LD $R^2$ greater than $R_T$) with another SNP having a more significant marginal $p$ value (or $Z$-score statistic value). For the remaining selected test SNPs, we use marginal standardized eQTL effect sizes from eQTL summary data as eQTL weights for TWAS in Stage II. We considered $R_T = 0.99$ and $P_T = (0.001, 0.05)$ in this paper and implemented the P+T method using PLINK 1.9[55] (see Code availability). We denote the P+T method with $P_T$ equal to 0.001 and 0.05 as P+T (0.001) and P+T (0.05), respectively.

Frequentist lassosum: with standardized $\mathbf{e}_g$ and $\mathbf{X}_g$, we can show that the marginal least squared eQTL effect size estimates from the single variant regression model (Eq. (3)) is $\widetilde{\mathbf{w}} = \mathbf{X}_g^\mathsf{T} \mathbf{e}_g / n$ and that the LD correlation matrix is $\mathbf{R} = \mathbf{X}_g^\mathsf{T} \mathbf{X}_g / n$. That is,

$$\mathbf{X}_g^\mathsf{T} \mathbf{e}_g = n\widetilde{\mathbf{w}} \text{ and } \mathbf{X}_g^\mathsf{T} \mathbf{X}_g = n\mathbf{R}. \quad (4)$$

By approximating $n\mathbf{R}$ by $n\mathbf{R}_s (\mathbf{R}_s = (1-s)\mathbf{R}_r + s\mathbf{I})$ with a tuning parameter $0 < s < 1$, a reference LD correlation matrix $\mathbf{R}_r$ from an external panel such as one from the 1000 Genomes Project[58], and an identity matrix $\mathbf{I}$) in the LASSO[32] penalized loss function, the frequentist lassosum method[27] can tune the LASSO penalty parameter and $s$ using a pseudovalidation approach and then solve for eQTL effect size estimates $\hat{\mathbf{w}}$ by minimizing the approximated LASSO loss function requiring no individual-level data (see details in Supplementary Methods).

Bayesian SDPR: Bayesian DPR method[33] as implemented in TIGAR[22] estimates $\hat{\mathbf{w}}$ for the underlying multivariable regression model in Eq. (1) by assuming a normal prior $N(0, \sigma_w^2)$ for $w_j$ and a Dirichlet process prior[59] $DP(H, \alpha)$ for $\sigma_w^2$ with base distribution $H$ and concentration parameter $\alpha$. SDPR[29] assumes the same DPR model but can be applied to estimate the eQTL effect sizes $\hat{\mathbf{w}}$ using only eQTL summary data (see details in Supplementary Methods).

Bayesian PRS-CS: the PRS-CS method[28] assumes the following normal prior for $w_j$ and non-informative scale-invariant Jeffreys prior on the residual variance $\sigma_\epsilon^2$ in Eq. (1):

$$w_j \sim N\left(0, \frac{\sigma_\epsilon^2}{n} \psi_j\right), p(\sigma_\epsilon^2) \propto \sigma_\epsilon^2, \psi_j \sim Gamma\left(a, \delta_j\right), \delta_j \sim Gamma(b, \phi),$$

where local shrinkage parameter $\psi_j$ has an independent gamma-gamma prior and $\phi$ is a global-shrinkage parameter controlling the overall sparsity of $\mathbf{w}$. PRS-CS sets hyper parameters $a = 1$ and b $= 1/2$ to ensure the prior density of $w_j$ to have a sharp peak around zero to shrink small effect sizes of potentially false eQTL towards zero, as well as heavy, Cauchy-like tails which assert little influence on eQTLs with larger effects. Posterior estimates $\hat{\mathbf{w}}$ will be obtained from eQTL summary data (i.e., marginal effect size estimates $\widetilde{\mathbf{w}}$ and $p$ values) and reference LD correlation matrix $\mathbf{R}$ by Gibbs Sampler (see details in Supplementary Methods). We set $\phi$ as the square of the proportion of causal variants in the simulation and as $10^{-4}$ per gene in the real data application.

## OTTERS framework

As shown in Fig. 1, OTTERS first trains GReX imputation models per gene $g$ using P+T, lassosum, SDPR, and PRS-CS methods that each infers cis-eQTLs weights using cis-eQTL summary data and an external LD reference panel of similar ancestry (Stage I). Once we derive cis-eQTLs weights for each training method, we can impute the respective GReX using that method and perform the respective gene-based association analysis in the test GWAS dataset using the formulas given in Eq. (2) (Stage II). We thus derive a set of TWAS $p$ values for gene $g$; one $p$ value for each training model that we applied. We then use these TWAS $p$ values to create an omnibus test using the ACAT-O[34] approach that employs a Cauchy distribution for inference (see details in Supplementary Methods). We refer to the $p$ value derived from ACAT-O test as the OTTERS $p$ value.

## Marginal eQTL effect sizes

In practice of training GReX imputation models using reference eQTL summary data, the marginal standardized eQTL effect sizes were approximated by $\widetilde{w}_j \approx Z_j / \sqrt{\text{median}(n_{g,j})}$, where $Z_j$ denotes the corresponding eQTL $Z$-score statistic value by single variant test and median$(n_{g,j})$ denotes the median sample size of all cis-eQTLs for the target gene $g$. The median cis-eQTL sample size per gene was also taken as the sample size value required by lassosum, SDPR, and PRS-CS methods, for robust performance. Since summary eQTL datasets (e.g., eQTLGen) were generally obtained by meta-analysis of multiple cohorts, the sample size per test SNP could vary across all cis-eQTLs of the test gene. The median cis-eQTL sample size ensures a robust performance for applying those eQTL summary data-based methods.

## LD clumping

We performed LD clumping with $R_T = 0.99$ for all individual methods in both simulation and real studies. Using PRS-CS as an example, we also showed that LD clumping does not affect the GReX imputation accuracy compared to no clumping in real data testing (Supplementary Fig. S12).

## LD blocks for lassosum, PRS-CS, and SDPR

LD blocks were determined externally by ldetect[60] for lassosum and PRS-CS, while internally for SDPR, which ensures that SNPs in one LD block do not have nonignorable correlation ($R^2 > 0.1$) with SNPs in other blocks.

## Simulate GWAS $Z$-score

Given gene expression $\mathbf{e}_g$ simulated from the multivariate regression model $\mathbf{e}_g = \mathbf{X}_g \mathbf{w} + \boldsymbol{\epsilon}_g$ with standardized genotype matrix $\mathbf{X}_g$ and

$\boldsymbol{\epsilon}_g \sim N\left(0, (1 - h_e^2)\mathbf{I}\right)$, we assume GWAS phenotype data of $n_{gwas}$ samples are simulated from the following linear regression model

$$\mathbf{y} = h_p\left(\mathbf{X}_g \mathbf{w}\right) + \boldsymbol{\epsilon}_p, \boldsymbol{\epsilon}_p \sim N(0, \mathbf{I}).$$

Conditioning on true genetic effect sizes, the GWAS $Z$-score test statistics of all test SNPs will follow a multivariate normal distribution, $MVN\left(\boldsymbol{\Sigma}_g \mathbf{w}\sqrt{n_{gwas}h_p^2}, \boldsymbol{\Sigma}_g\right)$, where $\boldsymbol{\Sigma}_g$ is the correlation matrix of the standardized genotype $\mathbf{X}_g$ from test samples, and $h_p^2$ denotes the amount of phenotypic variance explained by simulated GReX = $\mathbf{X}_g\mathbf{w}$[38]. Thus, for a given GWAS sample size, we can generate GWAS $Z$-score statistic values from this multivariate normal distribution.

## FUSION using individual-level reference data

To train GReX imputation models by FUSION with individual-level reference data, we trained Best Linear Unbiased Predictor model[61], Elastic-net regression[62], LASSO regression[32], and single best eQTL model as implemented in the FUSION tool (see Code availability). Default settings were used to train GReX imputation models by FUSION in our simulation and real studies. LASSO regression was performed only for genes with positively estimated expression heritability. The eQTL weights of the best-trained GReX imputation model will be used to conduct TWAS by FUSION.

## GTEx V8 dataset

GTEx V8 dataset (dbGaP phs000424.v8.p2) contains comprehensive profiling of WGS genotype data and RNA-sequencing (RNA-seq) transcriptomic data across 54 human tissue types of 838 donors. The GTEx V8 WGS genotype data of all samples were used to construct reference LD in our studies. The GTEx V6 RNA-seq data of whole blood samples were used to train GReX imputation models by FUSION, and the GTEx V8 RNA-seq data of additional whole blood samples ($n = 315$) were used to test GReX imputation accuracy in our studies. GTEx V8 RNA-seq data of all whole blood samples ($n = 574$) were also used as reference data for comparing the performance of OTTERS and FUSION.

## eQTLGen consortium dataset

The eQTLGen consortium[23] dataset was generated based on meta-analysis across 37 individual cohorts ($n = 31,684$) including GTEx V6 as a sub-cohort. eQTLGen samples consist of 25,482 blood (80.4%) and 6202 peripheral blood mononuclear cell (19.6%) samples. We considered SNPs with minor allele frequency > 0.01, Hardy–Weinberg $p$ value > 0.0001, call rate > 0.95, genotype imputation $r^2$ > 0.5 and observed in at least two cohorts[23]. We only considered cis-eQTL (within ±1 MB around gene transcription start sites) with a test sample size > 3000. As a result, we used cis-eQTL summary data of 16,699 genes from eQTLGen to train GReX imputation models for use in OTTERS in this study.

## UK Biobank GWAS data of cardiovascular disease

Summary-level GWAS data of Cardiovascular Disease from UKBB ($n = 459,324$, case fraction = 0.319)[35] were generated by BOLT-LMM based on the Bayesian linear mixed model per SNP[63] with assessment centered, sex, age, and squared age as covariates. Although BOLT-LMM was derived based on a quantitative trait model, it can be applied to analyze case–control traits and has a well-controlled false-positive rate when the trait is sufficiently balanced with a case fraction ≥10% and samples are of the same ancestry. The tested dichotomous cardiovascular disease phenotype includes a list of sub-phenotypes: hypertension, heart/cardiac problem, peripheral vascular disease, venous thromboembolic disease, stroke, transient ischemic attack (tia), subdural hemorrhage/hematoma, cerebral aneurysm, high cholesterol, and other venous/lymphatic diseases.

## Reporting summary

Further information on research design is available in the Nature Portfolio Reporting Summary linked to this article.

## Data availability

ROS/MAP/MSBB WGS data used in our simulation studies are available through Synapse with data access application (https://www.synapse.org/#!Synapse:syn10901595). The eQTLGen consortium data are available from the consortium portal website (https://www.eqtlgen.org). UK Biobank summary-level GWAS data are available through the Alkes Group (https://alkesgroup.broadinstitute.org/UKBB). Individual-level GTEx reference data are available through dbGap (Accession phs000424.v8.p2). Summary eQTL data of blood tissue in GTEx cohort are available from GTEx Portal (https://console.cloud.google.com/storage/browser/gtex-resources/GTEx_Analysis_v8_QTLs/GTEx_Analysis_v8_eQTL_all_associations). Significant genes from TWAS-hub are available from http://twas-hub.org. The summary eQTL weights of blood tissue generated by OTTERS (from eQTLGen data) and summary TWAS results generated by OTTERS for cardiovascular disease (from UK Biobank data) are available from Synapse (https://doi.org/10.7303/syn51009573).

## Code availability

Source code for OTTERS is available from https://github.com/daiqile96/OTTERS. All scripts used to generate intermediate or final data and figures are available from GitHub page https://github.com/daiqile96/OTTERS_paper and available in Zenodo with the identifier https://doi.org/10.5281/zenodo.7566827. Source code for ACAT is available from https://github.com/yaowuliu/ACAT. Source code for FUSION is available from http://gusevlab.org/projects/fusion. Source code for lassosum is available from https://github.com/tshmak/lassosum. Source code for PRS-CS is available from https://github.com/getian107/PRScs. Source code for SDPR is available from https://github.com/eldronzhou/SDPR. Plink version 1.9 is used and available at https://www.cog-genomics.org/plink/.

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

## Acknowledgements

The authors thank Dr Greg Gibson from Georgia Tech for his insightful comments and discussion that help the development and improve the quality of this manuscript. This work was supported by National Institutes of Health grant awards R35GM138313 (Q.D., J.Y.), RF1AG071170 (Q.D., M.P.E.), and Estonian Research Council Grant PUT (PRG1291) for T.E. NIH/NIA grants P30AG10161, R01AG15819, R01AG17917, R01AG30146, R01AG36836, R01AG56352, U01AG32984, U01AG46152, U01AG61356, the Illinois Department of Public Health, the Translational Genomics Research Institute support the generation of the ROS/MAP data led by A.S.B., P.L.D.J. and D.A.B. The Young Finns Study has been financially supported by the Academy of Finland: grants 322098, 286284, 134309 (Eye), 126925, 121584, 124282, 255381, 256474, 283115, 319060, 320297, 314389, 338395, 330809, and 104821, 129378 (Salve), 117797 (Gendi), and 141071 (Skidi), the Social Insurance Institution of Finland, Competitive State Research Financing of the Expert Responsibility area of Kuopio, Tampere and Turku University Hospitals (grant X51001), Juho Vainio Foundation, Paavo Nurmi Foundation, Finnish Foundation for Cardiovascular Research, Finnish Cultural Foundation, The Sigrid Juselius Foundation, Tampere Tuberculosis Foundation, Emil Aaltonen Foundation, Yrjö Jahnsson Foundation, Signe and Ane Gyllenberg Foundation, Diabetes Research Foundation of Finnish Diabetes Association, EU Horizon 2020 (grant 755320 for TAXINOMISIS and grant 848146 for To Aition), European Research Council (grant 742927 for MULTIEPIGEN project), Tampere University Hospital Supporting Foundation, and Finnish Society of Clinical Chemistry and the Cancer Foundation Finland. The funders had no role in study design, data collection and analysis, decision to publish, or preparation of the manuscript.

## Author contributions

Q.D. conducted data analysis and drafted the manuscript. J.Y. and M.P.E. conceptualized and led the project, and edited the manuscript. G.Z. and H.Z. consulted data analysis and edited the manuscript. U.V., L.F., A.B., A.T., T.L., O.R. and T.E. contributed reference eQTL summary data and edited the manuscript. eQTLGen Consortium contributed eQTLGen summary data.

## Competing interests

The authors declare no competing interests.

## Additional information

## eQTLGen Consortium

Mawussé Agbessi[14], Habibul Ahsan[15], Isabel Alves[14], Anand Kumar Andiappan[16], Wibowo Arindrarto[17], Philip Awadalla[14], Alexis Battle[9], Frank Beutner[18], Marc Jan Bonder[6,19], Dorret I. Boomsma[20], Mark W. Christiansen[21], Annique Claringbould[6,7], Patrick Deelen[6,7,22,23], Tõnu Esko[5], Marie-Julie Favé[14], Lude Franke[6,7], Timothy Frayling[24], Sina A. Gharib[21,25], Greg Gibson[26], Bastiaan T. Heijmans[17], Gibran Hemani[27], Rick Jansen[28], Mika Kähönen[29], Anette Kalnapenkis[5], Silva Kasela[5], Johannes Kettunen[30], Yungil Kim[8,31], Holger Kirsten[32], Peter Kovacs[33], Knut Krohn[34], Jaanika Kronberg[5], Viktorija Kukushkina[5], Zoltan Kutalik[35], Bernett Lee[16], Terho Lehtimäki[10], Markus Loeffler[32], Urko M. Marigorta[26,36,37], Hailang Mei[38], Lili Milani[5], Grant W. Montgomery[39], Martina Müller-Nurasyid[40,41,42], Matthias Nauck[43,44], Michel G. Nivard[45], Brenda Penninx[28], Markus Perola[46], Natalia Pervjakova[5], Brandon L. Pierce[15],

Joseph Powell[47], Holger Prokisch[48,49], Bruce M. Psaty[21,50], Olli T. Raitakari[11,12,13], Samuli Ripatti[51], Olaf Rotzschke[16], Sina Rüeger[35], Ashis Saha[8], Markus Scholz[32], Katharina Schramm[40,41], Ilkka Seppälä[10], Eline P. Slagboom[17], Coen D. A. Stehouwer[52], Michael Stumvoll[53], Patrick Sullivan[54], Peter A. C. 't Hoen[55], Alexander Teumer [9], Joachim Thiery[56], Lin Tong[15], Anke Tönjes[44], Jenny van Dongen[20], Maarten van Iterson[17], Joyce van Meurs[57], Jan H. Veldink[58], Joost Verlouw[57], Peter M. Visscher[39], Uwe Völker[59], Urmo Võsa [5], Harm-Jan Westra[6,7], Cisca Wijmenga[6], Hanieh Yaghootka[24,60,61], Jian Yang[39,62], Biao Zeng[26] & Futao Zhang[39]

[14]Computational Biology, Ontario Institute for Cancer Research, Toronto, ON, Canada. [15]Department of Public Health Sciences, University of Chicago, Chicago, IL, USA. [16]Singapore Immunology Network, Agency for Science Technology and Research, Singapore, Singapore. [17]Leiden University Medical Center, Leiden, The Netherlands. [18]Heart Center Leipzig, Universität Leipzig, Leipzig, Germany. [19]European Molecular Biology Laboratory, Genome Biology Unit, 69117 Heidelberg, Germany. [20]Department of Biological Psychology, Vrije Universiteit Amsterdam, Amsterdam, The Netherlands. [21]Cardiovascular Health Research Unit, University of Washington, Seattle, WA, USA. [22]Genomics Coordination Center, University Medical Centre Groningen, Groningen, The Netherlands. [23]Department of Genetics, University Medical Centre Utrecht, P.O. Box 85500, 3508 GA Utrecht, The Netherlands. [24]Genetics of Complex Traits, University of Exeter Medical School, Royal Devon & Exeter Hospital, Exeter, UK. [25]Department of Medicine, University of Washington, Seattle, WA, USA. [26]School of Biological Sciences, Georgia Tech, Atlanta, GA, USA. [27]MRC Integrative Epidemiology Unit, University of Bristol, Bristol, UK. [28]Amsterdam UMC, Vrije Universiteit Amsterdam, Amsterdam, The Netherlands. [29]Department of Clinical Physiology, Tampere University Hospital and Faculty of Medicine and Health Technology, Tampere University, Tampere, Finland. [30]University of Helsinki, Helsinki, Finland. [31]Genetics and Genomic Science Department, Icahn School of Medicine at Mount Sinai, New York, NY, USA. [32]Institut für Medizinische InformatiK, Statistik und Epidemiologie, LIFE – Leipzig Research Center for Civilization Diseases, Universität Leipzig, Leipzig, Germany. [33]IFB Adiposity Diseases, Universität Leipzig, Leipzig, Germany. [34]Interdisciplinary Center for Clinical Research, Faculty of Medicine, Universität Leipzig, Leipzig, Germany. [35]Lausanne University Hospital, Lausanne, Switzerland. [36]Integrative Genomics Lab, CIC bioGUNE, Bizkaia Science and Technology Park, Derio, Bizkaia, Basque Country, Spain. [37]IKERBASQUE, Basque Foundation for Science, Bilbao, Spain. [38]Department of Medical Statistics and Bioinformatics, Leiden University Medical Center, Leiden, The Netherlands. [39]Institute for Molecular Bioscience, University of Queensland, Brisbane, QLD, Australia. [40]Institute of Genetic Epidemiology, Helmholtz Zentrum München – German Research Center for Environmental Health, Neuherberg, Germany. [41]Department of Medicine I, University Hospital Munich, Ludwig Maximilian's University, München, Germany. [42]DZHK (German Centre for Cardiovascular Research), partner site Munich Heart Alliance, Munich, Germany. [43]Institute of Clinical Chemistry and Laboratory Medicine, Greifswald University Hospital, Greifswald, Germany. [44]German Center for Cardiovascular Research (partner site Greifswald), Greifswald, Germany. [45]Department of Biological Psychology, Faculty of Behaviour and Movement Sciences, VU, Amsterdam, The Netherlands. [46]National Institute for Health and Welfare, University of Helsinki, Helsinki, Finland. [47]Garvan Institute of Medical Research, Garvan-Weizmann Centre for Cellular Genomics, Sydney, NSW, Australia. [48]Institute of Neurogenomics, Helmholtz Zentrum München, Neuherberg, Germany. [49]Institute of Human Genetics, Technical University Munich, Munich, Germany. [50]Kaiser Permanente Washington Health Research Institute, Seattle, WA, USA. [51]Statistical and Translational Genetics, University of Helsinki, Helsinki, Finland. [52]Department of Internal Medicine and School for Cardiovascular Diseases (CARIM), Maastricht University Medical Center, Maastricht, The Netherlands. [53]Department of Medicine, Universität Leipzig, Leipzig, Germany. [54]Department of Medical Epidemiology and Biostatistics, Karolinska Institutet, Stockholm, Sweden. [55]Center for Molecular and Biomolecular Informatics, Radboud Institute for Molecular Life Sciences, Radboud University Medical Center Nijmegen, Nijmegen, The Netherlands. [56]Institute for Laboratory Medicine, LIFE – Leipzig Research Center for Civilization Diseases, Universität Leipzig, Leipzig, Germany. [57]Department of Internal Medicine, Erasmus Medical Centre, Rotterdam, The Netherlands. [58]UMC Utrecht Brain Center, University Medical Center Utrecht, Department of Neurology, Utrecht University, Utrecht, The Netherlands. [59]Interfaculty Institute for Genetics and Functional Genomics, University Medicine Greifswald, Greifswald, Germany. [60]School of Life Sciences, College of Liberal Arts and Science, University of Westminster, 115 New Cavendish Street, London, UK. [61]Division of Medical Sciences, Department of Health Sciences, Luleå University of Technology, Luleå, Sweden. [62]School of Life Sciences, Westlake University, Hangzhou, Zhejiang, China.

