## [Peer Review File · Nature Communications]

OTTERS: A powerful TWAS framework leveraging summary-level reference dataREVIEWER COMMENTS

Reviewer #1 (Remarks to the Author):

Review: OTTERS: A powerful TWAS framework leveraging summary-level reference data

Overview: Dai et al present OTTERS, a TWAS framework, to identify gene/trait associations from summary-level GWAS and functional data. Most TWAS pipelines/approaches rely on individual-level data to first build predictive models of gene expression from genetic data, and then integrate those models with downstream GWAS summary data to perform association testing. While this approach has worked well in practice, a limitation is that recent ultra-large-scale eQTL data (e.g., eQTLGen) has only released summary statistics, which limits the application of traditional penalized model fitting. OTTERS proposes to leverage polygenic risk-score prediction tools to fit predictive models using the eQTL summary data, rather than individual-level data, and integrate weights with downstream GWAS data. In order to combine the results across many possible predictive approaches, OTTERS also proposes an ACAT-based meta analysis. I found the approach interesting, and the results largely supporting claims, however I have some comments regarding simulation design.

Major comments:

1. The authors performed simulations to assess OTTERS alongside FUSION and various PRS approaches. However, I found it somewhat limited by the decision to use the genetic variation around a single gene, rather than sample a random gene region, for each simulation. Similarly, null simulations relied on a single instance of model fitting with permutation.
2. I think it would be helpful to understand the circumstances of when certain PRS approaches perform better, by diving a bit more into the eQTLGen results and analyzing if there are architectural patterns associated with model selection. For example, genes with greater cis-heritability, mean LDscore, etc. I understand that part of the benefit of OTTERS is to combine this approaches regardless of which performs best individually, but shedding some light on these characteristics may help better understand results.

Minor comments:

1. "Existing TWAS tools require individual-level eQTL reference data [...]". While this is largely true, there are MR and SMR related tools that require only summary-level data to make inferences about gene/trait relationships.

Reviewer #2 (Remarks to the Author):

Enclosed is a review of Dai et al's manuscript:

"OTTERS: A powerful TWAS framework leveraging summary-level reference data"

In this manuscript, the authors present a novel TWAS framework that is able to use eQTL summary statistics to train predictive models of gene expression by repurposing different methods to train polygenic scores. OTTERS then integrates these models with GWAS to detect genetic associations. The authors conducted several simulations to present various genetic architectures where each PGS method provides the most powerful detection of genetic association and conducts a TWAS of cardiovascular disease to report novel finding. Overall, the method is interesting and clever, from the math to the name!

However, I have some (minor) concerns about the evaluation of the method. My main concern, which

I believe is a considerably important one, is the lack of direct comparison to an expression prediction method that leverages individual-level genotypes in real data. The main premise of my concern is this: if traditional individual-level predictive models of expression at such large sample sizes enable more powerful TWAS than OTTERS, then (a) this should be reported and (b) a duty of eQTL consortia to provide pre-trained weights corresponding to their results. Some of the co-authors of this manuscript are lead investigators for these large initiatives. This is a common practice now, given recent initiatives like OmicsPred (<https://www.omicspred.org/>). I appreciate that the authors consider this comparison in simulations, but in my experience, real data is often not similar.

Nevertheless, the aggregation of methods that consider different architectures is specifically an aspect of this method that seems intuitively important.

I detail my comments below:

RESULTS

1. Line 168 – are these 4 PGS methods the best out of other available methods? Did the authors consider more recent methods like MegaPRS (10.1038/s41467-021-24485-y) or PUMAS (10.1186/s13059-021-02479-9)?
2. Line 196 – as I mention above, I appreciate this comparison back to FUSION in the simulations! These results are very promising!
3. This is my remaining concern with the Results presented. I agree that the OTTERS framework for model training is useful when individual level genotypes are not available in the eQTL dataset. But the authors do not compare OTTERS to a traditional individual-level TWAS model (e.g., PrediXcan/FUSION). If individual-level TWAS methods outperform OTTERS, it seems reasonable to expect large QTL consortia to provide pre-computed predictive models of gene expression for the public, along with their eQTL summary statistics. I urge the authors to consider including a comparison to individual-level TWAS models in the real data applications in datasets of equal sample size (using whatever method they prefer, at whatever sample sizes, etc). I do not believe computational cost should be an issue: on similar Linux shell as the authors presented, I was able to train an elastic net model using glmnet or a SuSiE model with 32,000 samples and 10,000 features in around 3 minutes. Just a handful of comparisons would be helpful here.

Response to Reviewers

We are grateful for the reviewers' helpful comments, which have enhanced the quality of our manuscript. In addition to addressing all reviewer comments, we updated our TWAS analyses based on eQTLGen summary statistics and UK Biobank¹ (UKBB) GWAS summary data of cardiovascular disease by matching strand flip SNPs between the two datasets. Such matching increased the number of test SNPs used in TWAS and resulted in the identification of more TWAS risk genes than in our initial submission.

Within the manuscript without tracking changes, we denote new or substantially revised material by a vertical line in the left margin.

REVIEWER COMMENTS

Reviewer #1 (Remarks to the Author):

Overview: Dai et al present OTTERS, a TWAS framework, to identify gene/trait associations from summary-level GWAS and functional data. Most TWAS pipelines/approaches rely on individual-level data to first build predictive models of gene expression from genetic data, and then integrate those models with downstream GWAS summary data to perform association testing. While this approach has worked well in practice, a limitation is that recent ultra-large-scale eQTL data (e.g., eQTLGen) has only released summary statistics, which limits the application of traditional penalized model fitting. OTTERS proposes to leverage polygenic risk-score prediction tools to fit predictive models using the eQTL summary data, rather than individual-level data, and integrate weights with downstream GWAS data. In order to combine the results across many possible predictive approaches, OTTERS also proposes an ACAT-based meta-analysis. I found the approach interesting, and the results largely supporting claims. However, I have some comments regarding simulation design.

We would like to thank the reviewer for these encouraging comments.

Major comments:

1. The authors performed simulations to assess OTTERS alongside FUSION and various PRS approaches. However, I found it somewhat limited by the decision to use the genetic variation around a single gene, rather than sample a random gene region, for each simulation. Similarly, null simulations relied on a single instance of model fitting with permutation.

We thank the reviewer for this suggestion. We substantially revised our simulation study based on real whole genome sequencing (WGS) data from the ROS/MAP cohort^{2,3} and MSBB study⁴. We divided 14,772 genes within our WGS dataset into five groups according to gene length, and randomly selected 100 genes from each group (500 genes in total). We randomly split the WGS dataset into 568 training (30%) and 1326 testing samples (70%), to mimic a relatively small sample size in the real reference panel for training gene expression imputation models. We considered scenarios with 2 different proportions of causal cis-eQTL, $p_{causal} = (0.001, 0.01)$, and 3 different proportions of gene expression variance explained by causal eQTL, $h_e^2 = (0.01, 0.05, 0.1)$. For each gene in each scenario, we simulated 10 replicates of gene expression. For each simulated gene expression, we generated 10 sets of GWAS Z-scores to perform a total of 50,000 TWAS simulations per scenario. Let n_{gwas} denotes the assumed GWAS sample size, and h_p^2 denotes the amount of phenotypic variance explained by simulated GReX. We set $h_p^2 = 0.025$ and considered $n_{gwas} = (200K, 300K, 400K, 500K)$ for scenarios with $h_e^2 = 0.01$, $n_{gwas} =$

(25K, 50K, 75K, 100K) for scenarios with $h_e^2 = 0.05$, and $n_{gwas} = (10K, 20K, 30K, 40K)$ for scenarios with $h_e^2 = 0.1$.

Based on the new simulation results with 500 randomly selected genes, we demonstrated that the Stage I training method with the optimal test R^2 and TWAS power depended on the underlying genetic architecture of gene expression (p_{causal}) as well as gene expression heritability (h_e^2) (Figure 2 in the manuscript, which is reproduced below). In situations where true cis-eQTLs were sparse ($p_{causal} = 0.001$) and the gene expression heritability was small ($h_e^2 = 0.01$), $P+T (0.05)$ method performed the best with the highest TWAS power among all individual methods. When gene expression heritability increased ($h_e^2 = 0.05$ or 0.1) within this sparse eQTL model, $P+T (0.001)$ and PRS-CS were generally the optimal methods. For a less sparse model with $p_{causal} = 0.01$, SDPR and PRS-CS generally performed best among the individual methods. Relative to individual methods, however, we found that combining the TWAS p-values based on the four PRS training methods together for analysis in our OTTERS framework obtained the highest power across all scenarios.

Figure 2: Test R^2 (A) and TWAS power (B) comparison in simulation studies

To evaluate the type I error, we picked one simulated replicate per gene from the scenario with $h_e^2 = 0.1$ and $p_{causal} = 0.001$, simulated 2×10^3 phenotypes from $N(0,1)$, and permuted the eQTL weights for TWAS to perform a total of 10^6 null simulations. OTTERS was well calibrated as shown by quantile-quantile (QQ) plots of TWAS p-values (Figure S1, which is reproduced below). We observed that OTTERS had well-controlled type I error for stringent significance levels between 10^{-4} and 2.5×10^{-6} (Table S1, which is reproduced below) that are typically utilized in TWAS. For more modest significance thresholds ($\alpha = 10^{-2}$), we noted that OTTERS had slightly inflated type-I error rate; this result is consistent with the findings of the original ACAT work (Liu et al. AJHG 104: 410)⁵ that showed the Cauchy-distribution-based approximation may not be accurate for larger p-values when correlation among tests is strong (see Appendix A of Liu et al). Since most TWAS analyses need to adjust for analysis of $\sim 20K$ genes and therefore consider significance levels between 10^{-5} and 10^{-6} , we do not believe this result to diminish the practical impact of OTTERS but we do suggest that users interpret more modest p-values with caution.

We have updated all simulation results in the Results section (pages 6-9, lines 152–220) in the revised manuscript.

Table S1: TWAS type I errors under null simulation studies with $h_e^2 = 0.1$ and $p_{causal} = 0.001$ by 5 individual methods $P+T(0.001)$, $P+T(0.05)$, *lassosum*, *SDPR*, *PRS-CS*, and *OTTERS*.

Significance Level	$P+T(0.001)$	$P+T(0.05)$	lassosum	SDPR	PRS-CS	OTTERS
1.0×10^{-2}	1.71×10^{-2}	1.25×10^{-2}	1.12×10^{-2}	1.08×10^{-2}	1.12×10^{-2}	1.60×10^{-2}
1.0×10^{-4}	7.13×10^{-5}	9.60×10^{-5}	1.32×10^{-4}	9.70×10^{-5}	1.13×10^{-4}	9.30×10^{-5}
2.5×10^{-6}	0	0	5.00×10^{-6}	5.00×10^{-6}	5.00×10^{-6}	4.00×10^{-6}

Figure S1: Quantile-Quantile plots for $P+T(0.001)$, $P+T(0.05)$, *lassosum*, *SDPR*, *PRS-CS*, and *OTTERS* under null simulations.

2. I think it would be helpful to understand the circumstances of when certain PRS approaches perform better, by diving a bit more into the eQTLGen results and analyzing if there are architectural patterns associated with model selection. For example, genes with greater cis-heritability, mean LDscore, etc. I understand that part of the benefit of *OTTERS* is to combine this approaches regardless of which performs best individually, but shedding some light on these characteristics may help better understand results.

We want to thank reviewer for this suggestion. To better understand the differences among individual methods, we plotted the eQTL weights estimated by $P+T(0.001)$, $P+T(0.05)$, *lassosum*, *SDPR*, and *PRS-CS* for three example genes that were only detected by one or two individual methods (Figures S5-S7, which are reproduced below). For these genes, we plotted the eQTL weights produced by each method with such weights color coded with respect to

$-\log_{10}$ (GWAS p-values) from the UKBB GWAS summary statistics and shape coded with respect to the direction of UKBB GWAS Z-score statistics. This was based on the logic that significant TWAS p-values should be obtained by methods that obtained eQTL weights with relatively large magnitude for SNPs with relatively more significant GWAS p-values.

In Figure S5, we show the eQTL weights for gene *SIDT2*, which was a significant risk gene identified by both *PRS-CS* and *SDPR*, and had p-values $< 10^{-4}$ by other methods. Compared to *lassosum*, *SDPR* had more significant GWAS SNPs colocalized with eQTLs having relatively large weights in the test region, and *PRS-CS* had more non-significant GWAS SNPs colocalized with eQTLs having zero weights. Compared to the P+T methods, *SDPR* and *PRS-CS* based on a multivariate regression model modeled LD among all test SNPs, and thus estimated eQTL weights leading to significant TWAS findings.

In Figure S6, we provide the results of the gene *EDN3*, which was only identified by *P+T* methods (p-values $\leq 9.15 \times 10^{-8}$). Compared to *P+T* methods, *SDPR* (p-value = 5.9×10^{-3}) and *PRS-CS* (p-value = 0.0158) had fewer significant GWAS SNPs colocalized with eQTLs that had relatively large weights in the test region, while *lassosum* (p-value = 8.6×10^{-6}) assigned relatively large weights to more non-significant GWAS SNPs.

In Figure S7, we provide results for gene *LINC01093*, which was only identified by *lassosum*. For this gene, *SDPR* and *PRS-CS* estimated near-zero weights for most SNPs with significant GWAS p-values in the test region. Most significant GWAS SNPs did not have eQTL test p-values < 0.001 or 0.05, and were thus filtered out by *P+T* methods. *lassosum* was the only method that produced relatively large eQTL weights that co-localized with GWAS significant SNPs.

We include these figures in the supplemental material (Figures S5-S7) and mention them in the Results section (page 13, lines 312-328).

Figure S5: Cis-eQTL weights estimated by individual methods for gene *SIDT2*, with color coded with respect to $-\log_{10}(\text{p-value})$ from the UKBB GWAS summary statistics and shape coded with respect to the direction of GWAS Z-score statistics.

Figure S6. Cis-eQTL weights estimated by individual methods for gene *EDN3*, with color coded with respect to $-\log_{10}(\text{p-value})$ from the UKBB GWAS summary statistics and shape coded with respect to the direction of GWAS Z-score statistics.

Figure S7. cis-eQTL weights estimated by individual methods for gene LINC01093, with color coded with respect to $-\log_{10}(\text{p-value})$ from the UKBB GWAS summary statistics and shape coded with respect to the direction of GWAS Z-score statistics.

Minor comments:

1. “Existing TWAS tools require individual-level eQTL reference data [...]”. While this is largely true, there are MR and SMR related tools that require only summary-level data to make inferences about gene/trait relationships.

We thank the reviewer for pointing out this sentence. We now discuss how SMR can be used as an alternative analysis method besides TWAS in the Discussion (page 16, lines 392 - 396).

Reviewer #2 (Remarks to the Author):

Enclosed is a review of Dai et al's manuscript:

“OTTERS: A powerful TWAS framework leveraging summary-level reference data”

In this manuscript, the authors present a novel TWAS framework that is able to use eQTL summary statistics to train predictive models of gene expression by repurposing different methods to train polygenic scores. OTTERS then integrates these models with GWAS to detect genetic associations. The authors conducted several simulations to present various genetic architectures where each PGS method provides the most powerful detection of genetic association and conducts a TWAS of cardiovascular disease to report novel finding. Overall, the method is interesting and clever, from the math to the name!

However, I have some (minor) concerns about the evaluation of the method. My main concern, which I believe is a considerably important one, is the lack of direct comparison to an expression prediction method that leverages individual-level genotypes in real data. The main premise of my concern is this: if traditional individual-level predictive models of expression at such large sample sizes enable more powerful TWAS than OTTERS, then (a) this should be reported and (b) a duty of eQTL consortia to provide pre-trained weights corresponding to their results. Some of the co-authors of this manuscript are lead investigators for these large initiatives. This is a common practice now, given recent initiatives like OmicsPred (<https://www.omicspred.org/>). I appreciate that the authors consider this comparison in simulations, but in my experience, real data is often not similar.

Nevertheless, the aggregation of methods that consider different architectures is specifically an aspect of this method that seems intuitively important.

We would like to thank the reviewer for the positive comments about OTTERS. Please see our detailed point-to-point responses as follows.

I detail my comments below:

RESULTS

1. Line 168 – are these 4 PGS methods the best out of other available methods? Did the authors consider more recent methods like MegaPRS (10.1038/s41467-021-24485-y) or PUMAS (10.1186/s13059-021-02479-9)?

We thank the reviewer for the comment. The main aim of our work was to show that PRS methods can be employed to perform TWAS using summary-level eQTL data and that combining the results of multiple PRS methods together in an optimal omnibus test using ACAT-O can be more powerful than simply using a single PRS method for this purpose. The PRS methods currently utilized in the OTTERS framework are representative of techniques that make different assumptions of the underlying eQTL architecture; thereby ensuring that OTTERS remains powerful across different generating mechanisms. We agree that more recent PRS methods could be added to the OTTERS framework, and could possibly yield improved results, but this additional integration is beyond the scope of this work. We now mention that MegaPRS⁶ and PUMAS⁷ can be included as additional methods for training gene imputation models and can be incorporated into OTTERS in the Discussion (page 16, lines 389–391).

2. Line 196 – as I mention above, I appreciate this comparison back to FUSION in the simulations! These results are very promising!

We appreciate the reviewer for recognizing the advantages of OTTERS in our simulation studies.

3. This is my remaining concern with the Results presented. I agree that the OTTERS framework for model training is useful when individual level genotypes are not available in the eQTL dataset. But the authors do not compare OTTERS to a traditional individual-level TWAS model (e.g., PrediXcan/FUSION). If individual-level TWAS methods outperform OTTERS, it seems reasonable to expect large QTL consortia to provide pre-computed predictive models of gene expression for the public, along with their eQTL summary statistics. I urge the authors to consider including a comparison to individual-level TWAS models in the real data applications in datasets of equal sample size (using whatever method they prefer, at whatever sample sizes, etc). I do not believe computational cost should be an issue: on similar Linux shell as the authors presented, I was able to train an elastic net model using glmnet or a SuSiE model with 32,000 samples and 10,000 features in around 3 minutes. Just a handful of comparisons would be helpful here.

We want to thank the reviewer for this suggestion, which we have addressed by running an additional TWAS of cardiovascular disease in the UKBB where, for reference, we used individual-level transcriptomic data available from 574 GTEx V8 whole blood samples. We trained FUSION on this individual data, and then compared results to OTTERS using cis-eQTL summary statistics from these same 574 GTEx V8 whole blood samples coupled to reference LD from GTEx V8 WGS samples. We first compared training R^2 of gene expression imputation models trained by FUSION with single summary-level method: $P+T$ (0.001), $P+T$ (0.05), *lassosum*, *SDPR*, and *PRS-CS*. The training R^2 is simply the squared Pearson correlation coefficient between imputed GReX and true gene expression in the 574 training samples. We considered trained imputation models with training $R^2 > 0.01$ as "valid" training models, according to previous TWAS methods^{8,9}. By comparing training R^2 per "valid" GReX imputation model by *lassosum* versus the other methods (Figure S10, which is reproduced below), we observed that *lassosum* had the best overall performance for imputing GReX as it provided the most "valid" models with higher training R^2 than FUSION and other single summary-level methods.

Next, we compared FUSION with OTTERS with respect to the TWAS results using the same GWAS summary data of UKBB. We tested 19,653 genes and identified genes with p-values $< 2.53 \times 10^{-6}$ (Bonferroni corrected significance level) as significant TWAS genes. To identify independently significant TWAS genes, we calculated the R^2 (squared correlation) of estimated genetic regulated gene expression (GReX) by *lassosum* for each pair of genes. For a pair of genes with estimated GReX $R^2 > 0.5$, we only kept the gene with the smaller TWAS p-value as the independently significant gene. OTTERS obtained 34 independently significant TWAS genes, while FUSION identified 21 significant TWAS genes (Figure S11 and Table S3, which are reproduced below). A total of 14 genes were identified by both FUSION and OTTERS. This comparison between FUSION and OTTERS using the same training and test data demonstrates that OTTERS finds more risk genes than FUSION even when individual-level reference data are available. This finding in real data is consistent with our simulation results.

We report these findings in the Results section (pages 14–15, lines 342–366) and in the supplemental materials (Figures S10-S11, Table S3). Based on these findings, we don't believe it is necessary for consortia to provide pre-computed predictive models of gene expression for the public. In any event, we feel the production of such models would likely face serious practical hurdles in creation. As consortia like eQTLGen are composed of multiple individual studies, the delivery of these results would require each individual study to run their own predictive model separately (each using the same procedure and QC pipeline) and then somehow combine the results across studies dealing with different sample sizes, different training R^2 , and different covariates. In the unlikely scenario a mega-analysis is possible where the individual data from multiple studies can be pooled together for prediction after approval of complicated data-transfer agreements, the potential heterogeneity among studies due to LD, ancestry, or other factors still could lead to inaccurate inference.

Figure S10. Training R^2 in GTEx V8 whole blood samples by *lassosum* versus $P+T(0.001)$, $P+T(0.05)$, $PRS-CS$, $SDPR$, and $FUSION$. Training R^2 by *lassosum* versus $P+T(0.001)$ (A), $P+T(0.05)$ (B), $PRS-CS$ (C), $SDPR$ (D), and $FUSION$ (E) with 574 GTEx V8 training samples, with different colors denoting whether the imputation $R^2 > 0.01$ only by *lassosum* (red), only by the y axis method (green), or both methods (blue). Genes with $R^2 \leq 0.01$ by both methods were excluded from the plot.

Figure S11. Manhattan plot of TWAS results by OTTERS(A) and FUSION(B) using eQTL summary statistics from GTex V8 whole blood samples. Independently significant TWAS risk genes are labeled.

Table S3. Independent TWAS risk genes of cardiovascular disease identified by OTTERS using eQTL summary statistics from GTEx V8 whole blood samples.

CHROM	GeneName	OTTERS	FUSION	P+T(0.001)	P+T(0.05)	lassosum	SDPR	PRS-CS
1	MTHFR	3.99E-08	1.12E-03	8.05E-09	1.11E-06	7.17E-05	1.06E-03	7.95E-05
1	NPPA	1.33E-08	6.35E-06	1.19E-08	3.52E-09	1.39E-07	5.71E-04	5.94E-04
1	ZDHHC18	1.02E-07	7.35E-07	1.82E-07	5.27E-07	2.67E-08	9.34E-07	3.18E-07
1	PSRC1	8.75E-11	1.70E-04	1.91E-11	1.61E-06	1.23E-09	3.11E-10	1.45E-09
4	OR7E94P	2.14E-31	1.16E-23	4.28E-32	2.53E-04	2.16E-10	2.80E-05	1.35E-11
4	USP38	3.98E-07	4.99E-07	2.85E-05	1.09E-04	8.12E-08	7.85E-06	1.41E-05
7	AC005532.5	2.26E-06	1.03E-03	4.56E-07	4.88E-05	3.89E-04	1.17E-03	7.48E-04
7	GATSL2	5.12E-07	1.69E-02	7.59E-07	1.18E-07	1.70E-03	3.45E-03	7.02E-04
10	NT5C2	1.09E-08	2.42E-04	2.69E-09	1.76E-06	1.17E-08	1.03E-03	6.62E-05
11	TNNT3	4.89E-07	1.65E-05	1.01E-07	1.16E-05	7.80E-06	2.72E-05	3.02E-05
11	MRPL23-AS1	1.13E-08	2.48E-01	2.26E-09	1.21E-05	8.89E-05	7.11E-02	8.61E-04
11	KCNQ1OT1	2.47E-06	7.37E-03	5.09E-07	1.32E-04	1.80E-05	1.85E-02	4.39E-03
11	DDB2	1.46E-07	2.14E-04	1.00E-07	3.52E-06	4.17E-08	4.10E-04	1.18E-03
11	SIPA1	3.42E-07	4.77E-06	1.24E-07	3.49E-05	1.72E-07	4.31E-05	1.45E-06
11	DRAP1	2.46E-06	9.41E-02	4.92E-07	4.16E-02	1.88E-01	4.20E-02	9.94E-02
11	ZPR1	2.64E-08	3.11E-07	9.47E-07	5.34E-09	2.06E-05	3.44E-06	1.09E-06
12	POC1B	6.76E-08	8.87E-05	1.35E-08	5.55E-05	1.19E-05	9.13E-04	1.08E-04
12	FAM109A	4.25E-07	1.44E-05	3.58E-07	3.05E-07	1.04E-03	4.69E-07	2.82E-07
12	SH2B3	4.27E-07	1.37E-07	2.01E-07	5.86E-07	4.36E-04	9.32E-07	2.52E-07
15	CSK	1.18E-08	9.12E-07	3.12E-07	4.20E-07	5.53E-01	2.28E-08	2.68E-09
15	ULK3	1.18E-08	1.05E-06	8.28E-09	1.05E-07	6.39E-01	4.27E-09	1.68E-08
15	MPI	2.08E-09	2.69E-07	3.61E-08	1.58E-07	4.47E-01	6.20E-10	1.32E-09
15	RP11-69H7.2	9.54E-08	5.12E-03	1.91E-08	2.66E-05	1.39E-04	6.35E-05	2.78E-05
15	FES	2.61E-13	8.18E-09	8.34E-14	1.16E-08	4.48E-12	4.35E-12	1.48E-13
18	C18orf8	5.71E-07	2.32E-06	9.16E-06	1.87E-06	5.94E-07	1.99E-07	7.05E-07
19	CARM1	7.43E-08	7.53E-02	3.12E-06	1.49E-08	6.60E-04	2.52E-03	1.87E-03
19	SMARCA4	6.24E-07	5.14E-07	4.26E-07	9.35E-07	2.17E-07	1.34E-01	1.37E-03
19	HAUS8	9.29E-10	1.20E-06	1.56E-09	1.85E-09	2.38E-10	5.11E-07	9.97E-08
19	MYO9B	1.04E-06	8.26E-05	2.15E-07	9.87E-06	4.28E-04	3.78E-03	2.97E-05
19	NTN5	7.77E-09	9.62E-06	1.55E-09	3.95E-06	5.29E-06	4.78E-03	2.56E-04
20	PPP1R3D	2.63E-08	1.09E-01	3.21E-05	5.25E-09	3.51E-01	9.16E-04	8.22E-03
20	OPRL1	1.74E-10	1.28E-07	4.16E-08	8.25E-07	3.53E-11	5.25E-09	4.55E-09
20	LKAAEAR1	7.50E-07	3.00E-01	1.50E-07	8.47E-04	2.68E-01	1.14E-02	1.04E-03
21	RRP1B	2.35E-07	2.00E-06	5.73E-08	4.56E-07	6.27E-07	8.62E-03	1.32E-04

References

1. Loh, P.-R., Kichaev, G., Gazal, S., Schoech, A. P. & Price, A. L. Mixed-model association for biobank-scale datasets. *Nat Genet* **50**, 906–908 (2018).
2. Bennett, D. A. *et al.* Religious Orders Study and Rush Memory and Aging Project. *J Alzheimers Dis* **64**, S161–S189 (2018).
3. Bennett, D. A., Schneider, J. A., Arvanitakis, Z. & Wilson, R. S. OVERVIEW AND FINDINGS FROM THE RELIGIOUS ORDERS STUDY. *Curr Alzheimer Res* **9**, 628–645 (2012).
4. Wang, M. *et al.* The Mount Sinai cohort of large-scale genomic, transcriptomic and proteomic data in Alzheimer's disease. *Sci Data* **5**, 180185 (2018).
5. Liu, Y. *et al.* ACAT: A Fast and Powerful p Value Combination Method for Rare-Variant Analysis in Sequencing Studies. *The American Journal of Human Genetics* **104**, 410–421 (2019).
6. Zhang, Q., Privé, F., Vilhjálmsson, B. & Speed, D. Improved genetic prediction of complex traits from individual-level data or summary statistics. *Nat Commun* **12**, 4192 (2021).
7. Zhao, Z. *et al.* PUMAS: fine-tuning polygenic risk scores with GWAS summary statistics. *Genome Biology* **22**, 257 (2021).
8. Bhattacharya, A., Li, Y. & Love, M. I. MOSTWAS: Multi-Omic Strategies for Transcriptome-Wide Association Studies. *PLoS Genetics* **17**, e1009398 (2021).
9. Gusev, A. *et al.* Integrative approaches for large-scale transcriptome-wide association studies. *Nat Genet* **48**, 245–252 (2016).

REVIEWERS' COMMENTS

Reviewer #1 (Remarks to the Author):

The authors have performed a great deal of work to address my previous comments, and I applaud them for their efforts. I have no new comments at this point.

Reviewer #2 (Remarks to the Author):

I commend the authors for their thorough comparisons to address my remaining comments.

Response to Reviewers

REVIEWERS' COMMENTS

Reviewer #1 (Remarks to the Author):

The authors have performed a great deal of work to address my previous comments, and I applaud them for their efforts. I have no new comments at this point.

Reviewer #2 (Remarks to the Author):

I commend the authors for their thorough comparisons to address my remaining comments.

Response: We are grateful for the reviewers' helpful comments, which have enhanced the quality of our manuscript. We are very glad that both reviewers found our last revision and responses satisfactory.